# Water vapor anomaly over the tropical western Pacific in El Niño winters from radiosonde and satellite observations and ERA5 reanalysis data

Minkang Du[1,2,3]   Kaiming Huang[1,2,3]   Shaodong Zhang[1,2]   Chunming Huang[1,2]

Yun Gong[1,2]   and   Fan Yi[1,2,3]

[1]School of Electronic Information, Wuhan University, Wuhan, China

[2]Key Laboratory of Geospace Environment and Geodesy, Ministry of Education, Wuhan, China

[3]State Observatory for Atmospheric Remote Sensing, Wuhan, China

Correspondence: Kai Ming Huang (hkm@whu.edu.cn)

**Abstract.** Using radiosonde observations at five stations in the tropical western Pacific and reanalysis data for 15 years from 2005 to 2019, we report an extremely negative anomaly in atmospheric water vapor during the super El Niño winter of 2015/16, and compare the anomaly with that in the other three El Niño winters. Strong specific humidity anomaly is concentrated below 8 km of the troposphere with a peak at 2.5-3.5 km, and column integrated water vapor mass anomaly over the five radiosonde sites has a large negative correlation coefficient of -0.63 with oceanic Niño3.4 index, but with a lag of about 2-3 months. In general, the tropical circulation anomaly in the El Niño winter is characterized by divergence (convergence) in the lower troposphere over the tropical western (eastern) Pacific, thus the water vapor decreases over the tropical western Pacific as upward motion is suppressed. The variability of the Hadley circulation is quite small and has little influence on the observed water vapor anomaly. The anomaly of the Walker circulation makes a considerable contribution to the total anomaly in all the four El Niño winters, especially in the 2006/07 and 2015/16 eastern-Pacific (EP) El Niño events. The monsoon circulation shows a remarkable change from one event to another, and its anomaly is large in the 2009/10 and 2018/19 central-Pacific (CP) El Niño winters and small in the two EP El Niño winters. The observed water vapor anomaly is caused mainly by the Walker circulation anomaly in the super EP event of 2015/16 but by the monsoon circulation anomaly in the strong CP event of 2009/10. The roles of the Hadley, Walker and monsoon circulations in the EP and CP events are confirmed by the composite EP and CP El Niños based on the reanalysis data for 41 years. Owing to the anomalous decrease in upward transport of water vapor during the El Niño winter, lower cloud amounts and more outgoing longwave radiation over the five stations are clearly presented in satellite observation. In addition, a detailed comparison of water vapor in the reanalysis, radiosonde and satellite data shows a fine confidence level of the datasets, nevertheless, the reanalysis seems to slightly underestimate the water vapor over the five stations in the 2009/10 winter.

**1 Introduction**

As a dominant greenhouse gas in the atmosphere, water vapor has a profound impact on global

energy budgets not only through latent heat release upon phase transitions (Held and Soden, 2000), but

also through cloud formation that reflects long-wave radiation from below and short-wave radiation from

above (Stevens et al., 2017), thus water vapor plays a substantial role in the formation and evolution of

the climate system. The tropical Pacific is a major convection center and a region with abundant water

vapor. Sea surface temperature (SST) anomalies in the tropical Pacific has an important influence on

water vapor transport, cloud cover and precipitation distribution due to the tropical circulation changes

caused by El Niño-Southern Oscillation (ENSO). ENSO is characterized by anomalous SST in the

tropical Pacific. During ENSO, there is significant precipitation variability in the Euro-Mediterranean

(López-Parages and Rodríguez-Fonseca, 2012), Middle East (Sandeep and Ajayamohan, 2018),

southwest central Asia (Mariotti, 2007), western Africa (Okazaki et al., 2015), Pacific Ocean (Quartly et

al., 2000) and continental USA (Lee et al., 2014). ENSO has an effect on seasonal rainfall in East Asian

by inducing a weaker and later onset of the Indian monsoon circulation (Dai and Wigley, 2000; Zhao et

al., 2010; Yan et al., 2018). Vertical cloud anomalies in the tropical Atlantic from Aqua Moderate

Resolution Imaging Spectroradiometer are linked to ENSO-induced shift and weakening of the Walker

circulation and Hadley cell near the equator (Madenach et al., 2019). The strong 1997/98 El Niño

resulted in cloud structure anomalies and their radiative property changes over the tropical Pacific (Sun et

al., 2012), and increased upper tropospheric cirrus over the mid-Pacific but decreased cirrus over

Indonesia (Massie et al., 2000). Numerical investigation also indicated that warm water volume transport

and precipitation change are associated with ENSO (Ishida et al., 2008; Hill et al., 2009).

El Niño is generally classified into central-Pacific (CP) El Niño, also known as El Niño Modoki, and

eastern-Pacific (EP) El Niño based on distinct spatial distributions of warming SST anomaly averaged

over the Niño4 and Niño3 regions (Ashok et al., 2007; Yu and Kao, 2009; Yeh et al., 2009), respectively.

The 2006/07 and 2015/16 events are the EP El Niño because of the stronger SST anomaly during the

boreal winter (December to February, as DJF) in the Niño3 region than in the Niño4 region, while

correspondingly, the 2009/10 and 2018/19 events are categorized as the CP El Niño (Yeh et al., 2009).

The two types of El Niño have different effects on precipitation, surface temperature, moisture transport

and carbon cycle over many parts of the world (Weng et al., 2008; Kug et al., 2009; Wang et al., 2013;

Yeh et al., 2014; Gu and Adler, 2016; Wang et al., 2018). Su and Jiang (2013) and Takahashi et al. (2013)

suggested that water vapor anomaly over the tropical ocean is mainly controlled by thermodynamic

process during the 2006/07 EP El Niño, but by both dynamic and thermodynamic processes during the

2009/10 CP El Niño.

The EP El Niño in 2015/16 winter is one of the strongest ENSO events on record. Compared to the

strong 1982/83 and 1997/98 El Niños, the 2015/16 El Niño shows distinct aspects that the largest SST

anomalies are extended toward the central Pacific (Paek et al., 2017; L'Heureux et al., 2017). As the

unusual characteristics, the global effects of the 2015/16 event have attracted much attention. Palmeiro et

al. (2017) proposed that an early stratospheric final warming over the polar region and anomalous

precipitation over southern Europe in 2016 were related to the 2015/16 super El Niño. Li et al. (2018)

revealed that the combined effect of the 2015 ENSO warm phase and Madden-Julian Oscillation

(MJO)-4 index negative phase caused a significant deficit of precipitation on the Canadian Prairies in

May and June 2015. A striking freshwater anomaly was observed in the equatorial Pacific during the

onset of 2015/16 event (Gasparin and Roemmich, 2016), and rainfall $\delta^{18}O$ in the southern Papua was

generally enriched by 1.6‰–2‰ during the 2015 El Niño than during the 2013/14 ENSO-normal period

(Permana et al., 2016). Owing to convection anomaly during the 2015/16 El Niño, water vapor in the tropical lower stratosphere was increased by hydration of the lower stratosphere through convectively detrained cloud ice (Avery et al., 2017), and quasi-biennial oscillation in the tropical stratospheric wind was disrupted because of dramatic relocation of deep convection (Dunkerton, 2016; Newman et al., 2016). Hence, the 2015/16 El Niño had important influences on the circulation and composition transport and the mass exchange between the troposphere and stratosphere. In this paper, we investigate water vapor anomaly over the tropical western Pacific in the CP and EP El Niño events from radiosonde and satellite observations, in particular, extreme anomaly in the 2015/16 super El Niño winter, and explore the contributions of the tropical Hadley, Walker and monsoon circulation changes to the observed water vapor anomalies in the different El Niño events.

The data used are briefly described in section 2. In section 3, water vapor anomalies in four El Niño winters are presented, and the relationship between the ENSO intensity and the water vapor anomaly at the observational stations is explored. In section 4, we decompose the tropical circulation into the Hadley, Walker and monsoon circulation components, and estimate the roles of these circulations in the water vapor variation. Tropical cloud and outgoing longwave radiation (OLR) are investigated in section 5. A discussion of the water vapor data quality is provided in section 6. Finally, we summarize the results in section 7.

**2. Data**

In present study, we investigate the atmospheric water vapor by using radiosonde observations at five tropical stations for 15 years from January 2005 to December 2019, which are provided by the national oceanic and atmosphere administration (NOAA) at the website of

The five radiosonde stations are at Koror (7.33°N,
134.48°E), Yap (9.48°N, 138.08°E), Guam (13.55°N, 144.83°E), Truk (7.47°N, 151.85°E) and Ponape
(6.97°N, 158.22°E), located in the western Pacific warm pool. Balloon was launched twice daily at 0000
UT and 1200 UT, and during balloon ascent, sensing payload on balloon can obtain many meteorological
parameters, such as atmospheric pressure, temperature, relative humidity, and wind speed and direction.
We plot daily temperature, relative humidity, and wind speed time series observed by radiosonde to
identify potential outliers, and then the high resistant asymmetric biweight technique is applied to weed
out the outliers (Lanzante, 1996). The outlier data are very few, and the outliers of temperature, wind and
relative humidity account for only 0.09%, 0.08% and 0.02% of all observational data at the five stations
during 15 years, respectively. The radiosonde data is linearly interpolated to a vertical grid of 50 m, and
the interpolated data below 10 km is utilized to analyze the atmospheric water vapor variation. Burst
height of balloon is usually more than 30 km, thus the data availability below 10 km is high. In the period
that we focus on, the data are missing for about 4, 2, 1 and 4 months over Yap, Guam, Truk and Ponape,
respectively, and they are almost entirely from the several continuous observational missing rather than
balloon burst below 10 km.

Specific humidity can be derived from the profile of meteorological parameters observed by

radiosonde. The saturated vapor pressure $e_s$ is calculated according to a modified version of the
Magnus formula as follows (Murray, 1967),
$$e_s = 6.1078 \times \exp\left[\frac{17.269\left(T-273.16\right)}{T-35.86}\right] \tag{1}$$

where $T$ is the temperature in units of K. And then, the specific humidity $q$ (g kg$^{-1}$) is determined
from the following equations,

$$e = RH \times e_s \qquad (2)$$

$$q = \frac{0.622e}{p - 0.378e} \qquad (3)$$

where $e$ is the vapor pressure; $RH$ is the relative humidity; and $p$ is the pressure with units of hPa.

In addition, we use the monthly specific humidity, horizontal winds from surface to 300 hPa during the period of 2005-2019, obtained from the European centre for medium-range weather forecasts (ECMWF) ERA5 reanalysis data, to investigate the water vapor anomaly and tropical atmospheric circulation in the region of the radiosonde stations. The reanalysis data is produced by a sequential 4D variational data assimilation scheme, with a latitudinal and longitudinal resolution of $0.25° \times 0.25°$ at 37 pressure levels from 1000 to 1 hPa (Hersbach et al., 2020). The data is available at the website of https://cds.climate.copernicus.eu/cdsapp#!/home/.

To assess the atmospheric water vapor as compare to the reanalysis data and the radiosonde observations, a further evaluation is carried out using Aqua atmospheric infrared sounder (AIRS) water vapor mass mixing ratio data from 2005-2019. AIRS is a hyperspectral infrared spectrometer orbiting on the national aeronautics and space administration (NASA) Aqua spacecraft launched in May 2002, which can provide accurate measurements of temperature, moisture, and other atmospheric variables (Aumann et al., 2003). The data used here is water vapor vertical profiles from Level 3 monthly standard gridded retrieval product version 6, AIRS3STM (Susskind et al., 2014), which is available at http://disc.sci.gsfc.nasa.gov. The water vapor data contains 8 levels from 1000 and 300 hPa with a latitudinal and longitudinal grid of $1° \times 1°$, derived from the average of twice observations in two orbital overpasses per day. The ascending and descending orbits have equatorial crossing time at 13:30 local time (LT) and 1:30 LT, respectively.

Oceanic Niño index (ONI) is applied to discuss the correlation between the ENSO and the observed

water vapor anomaly. ONI is the measurement of ENSO strength, which is provided by the NOAA at
https://catalog.data.gov/dataset/climate-prediction-center-cpcoceanic-nino-index/. The ONI is defined as
a 3-month moving average of extended reconstructed sea surface temperature (ERSST) V5 sea surface
temperature anomalies in the Niño3.4 region at 5°N-5°S and 120°-170°W (Huang et al., 2017).
Cloud occurrence probability and OLR flux are also examined since they are sensitive to water vapor
variation (Stevens et al., 2017; Soden et al. 2008). The OLR data is measured by the NOAA-18 satellite,
which travel in sun-synchronous orbit with a 13:55 LT equatorial crossing time (Kramer, 2002). We use
the monthly OLR data between 2005 and 2019 from the NOAA archives with a latitudinal and
longitudinal grid of $2.5° \times 2.5°$ (Liebmann and Smith, 1996), which can be accessed through the website
of https://www.esrl.noaa.gov/psd/data/gridded/data.interp_OLR.html/. Cloud-aerosols lidar and infrared
pathfinder satellite observations (CALIPSO) are able to clearly identify cloud vertical structure (Winker
et al., 2007). The satellite has a sun-synchronous orbit with an equatorial crossing time around
1:30/13:30 LT (Stephens et al., 2002). Here, we use the CALIPSO Version 1.00 lidar level 3 cloud
occurrence monthly data in a latitudinal and longitudinal grid of $2° \times 2.5°$ with an altitude resolution of
60 m above the mean sea level, and the available data is from June 2006 to December 2016, downloaded
from the website of the NASA at https://eosweb.larc.nasa.gov/project/calipso/cloud_occurrence_table/.

**3 Water Vapor Anomaly**
**3.1 Water Vapor Anomaly during El Niño Winter**
We derive the profile of specific humidity from the radiosonde observations according to Eqs. (1-3),
and then calculate the monthly mean specific humidity. The monthly mean specific humidities in all the
same months are further averaged to obtain the monthly climatic normal, thus the monthly mean water
vapor anomaly is determined from the monthly mean series by subtracting the corresponding month
climatic normal. Figure 1 shows the monthly mean specific humidity anomaly based on the radiosonde
observations at Koror, Yap, Guam, Truk and Ponape from January 2005 to December 2019. Atmospheric
water vapor is mainly concentrated below 8 km, thus the large water vapor anomaly also occurs below 8
km. It can be seen from Fig. 1 that the observed water vapor anomaly is remarkably negative over the
five stations in the super El Niño winter of 2015-2016. The negative anomaly in the water vapor reaches
the peak values of -2.06 g kg$^{-1}$ around 3 km in January at Koror, -3.2 g kg$^{-1}$ around 3 km in February at
Yap, -2.39 g kg$^{-1}$ around 2.5 km in January at Guam, -2.29 g kg$^{-1}$ around 3.5 km in February at Truk and
-2.66 g kg$^{-1}$ around 2.5 km in February at Ponape, respectively. In the 2006/07, 2009/10 and 2018/19 El
Niño winters, the observed water vapor also exhibits the negative anomalies in the lower and middle
troposphere. We derive the monthly mean specific humidity anomaly from the reanalysis data at the
radiosonde stations during the same period, which is also presented in Fig. 1. The ERA5 reanalysis
shows water vapor anomaly scenario similar to the radiosonde observation. The negative anomalies in
the four El Niño winters are obvious in the reanalysis data, especially the strong anomaly in the 2015/16
event. Hence, the El Niño events can lead to the obvious reduction of water vapor in the region.
With the help of the ERA5 reanalysis data, we investigate the distribution of the abnormal water vapor
during the four El Niño events. Here, we introduce an important scalar of column integrated water vapor
mass (CWV), also called precipitable water, which is expressed as (Viswanadham, 1981),
$$Q = \frac{1}{g} \int_{P_z}^{P_0} q \, dp \tag{4}$$

where $Q$ is the CWV in units of kg m$^{-2}$; $g = 9.8$ m s$^{-2}$ is the acceleration due to gravity; and the
pressures $p_0$ and $p_z$ denote the bounds of integration, respectively. Considering that atmospheric
water vapor is mainly distributed below 8 km in the tropics due to the rapid decrease of water vapor with
height (Mapes et al., 2017), we choose $p_0 = 1000 \text{ hPa}$ on the ground and $p_z = 300 \text{ hPa}$ corresponding
to a height of about 9 km. According to Eq. (4), we calculate the CWV between 30°S and 30°N from
January 2005 to December 2019 based on the reanalysis data. Similarly, the monthly mean CWV and its
anomaly can be derived from the CWV series. Figure 2 presents the mean CWV anomalies in the four El
Niño winters. In the 2006/07 and 2015/16 EP El Niño events, the positive CWV anomalies appear in the
equatorial central and eastern Pacific, while in 2009/10 and 2018/19 CP El Niño events, the positive
anomalies concentrate in the central Pacific. This is consistent with previous studies (Kug et al., 2009;
Takahashi et al., 2013; Xu et al., 2017). The negative anomalies occur in the tropical western Pacific and
some tropical latitudes off the equator in both hemispheres. In the region of the five radiosonde stations,
the CWV anomaly is evidently negative and comparable between the 2009/10 and 2015/16 events
although the two events are classified into different El Niño types. Whereas in the other two events, the
water vapor anomaly is weak, which is in rough agreement with the radiosonde observation in Fig. 1.
**3.2 Relation between CWV Anomaly and ONI**
We choose the reanalysis CWV anomalies at the five radiosonde stations to discuss the relationship
between the water vapor anomaly and the ENSO. The monthly mean CWV anomaly averaged at the five
stations is derived from the radiosonde and reanalysis data from January 2005 to December 2019.
Considering that the ONI is a 3-month smoothed value, the monthly mean CWV anomaly is also
smoothed in a 3-month moving window. Figure 3 depicts the ONI and monthly mean CWV anomalies
from the radiosonde and reanalysis data. The CWV anomalies show a similar temporal evolution
between the observation and the reanalysis with a significant correlation coefficient $R$=0.83, but a
negative correlation to the ONI with a delay of about several months. The correlation coefficient between
the CWV anomaly and the ONI is calculated to be -0.63 (-0.62) with a lag of 3 (2) months. One can note

from Fig. 3 that when a strong La Niña occurs with ONI=-1.64 in November 2010, the water vapor

anomaly reaches the positive maximum in February and March 2011 from the observation and reanalysis

data, respectively. However, for the 2015/16 super El Niño event with the peak of ONI=2.6 in December

2015, an extremely negative anomaly appears in both the observation and reanalysis. The negative

anomaly attains as large as -5.39 and -5.75 kg m$^{-2}$ in February 2016 from the radiosonde and reanalysis

data, respectively. Similarly, the 2009/10 event has a large index of ONI=1.6 in November 2009, which

leads to the strong CWV anomalies of -2.45 and -3.94 kg m$^{-2}$ in January 2010 from the radiosonde and

reanalysis data, respectively. Hence, the ENSO or SST anomaly plays an important role in the water

vapor variation in the tropical western Pacific.

**4 Contribution from Tropical Circulations**

**4.1 Tropical Atmospheric Circulations**

Besides the SST effect, evaporated sea water is carried to higher levels by the upward flow, thus the

water vapor variability in the troposphere is closely related to the atmospheric circulation. In the tropics,

there are several well-known circulations, i.e. Hadley, Walker and monsoon circulations, and each

circulation has its own features and driving force though these circulations may be highly coupled with

each other. In this way, we attempt to estimate the contributions of each tropical circulation to the

observed water vapor anomalies in the El Niño events. According to the Helmholtz's theorem, horizontal

wind velocity can be decomposed into the rotational and divergent winds,

$$\vec{V_H} = \vec{V_\Psi} + \vec{V_\Phi} = \vec{k} \times \nabla\Psi - \nabla\Phi \tag{5}$$

where $\Psi$ is the stream function, $\Phi$ is the velocity potential; $\vec{k}$ is the unit vector in the vertical

direction; and $\vec{V}_H$, $\vec{V}_\Psi$ and $\vec{V}_\Phi$ are the horizontal, rotational and divergent wind velocities,

respectively. Thermal driving force resulted from differential heating and temperature contrast is
essential to cause atmospheric convergence-divergence and vertical motion and then the formation of
atmospheric circulation. The stream function involved in the rotation field has no contribution to the
atmospheric vertical motion, while the velocity potential may be chosen as the indicator of the
atmospheric circulations since it is in connection with the atmospheric convergence-divergence
associated with the upward and downward motions in the tropical region (Kanamitsu and Krishnamurti,
1978; Newell et al., 1996; Wang, 2002). Because atmospheric water vapor comes mainly from the lower
atmosphere through transport of ascending flow, we selected the velocity potential at 850 hPa to
represent the characteristics of the tropical circulations in the lower troposphere since the pressure level
was extensively used to investigate the lower atmospheric circulation (Wang, 2002; Weng et al., 2008;
Zhao et al., 2010). The divergence and velocity potential fields are calculated by using the ECMWF
reanalysis horizontal winds at 850 hPa according to the following equation (Krishnamurti, 1971; Tanaka
et al., 2004),

$$D = \nabla \cdot \vec{V}_H = -\nabla^2 \Phi \tag{6}$$

where $D$ is the divergence of horizontal wind. In Eq. (6), the negative sign means that the divergent wind
flows from the large velocity potential to the small velocity potential.
Based on the different driving mechanisms and movement features, Tanaka et al. (2004)
decomposed the tropical circulation in the upper troposphere (200 hPa) into the Hadley, Walker and
monsoon circulations, which have an advantage to quantitatively evaluate the intensity of the three
tropical circulations by means of the separation of the velocity potential into three orthogonal spatial
patterns. Subsequently, Takemoto and Tanaka (2007) used these circulation definitions to analyze the
Hadley, Walker, and monsoon circulations at 850 hPa of the lower troposphere, and compared the three
circulation components with those in the upper troposphere (200 hPa), which indicated that the velocity
potential intensities could be an index of each circulation in the lower troposphere without a notable
influence from the surface. Considering that atmospheric water vapor is mainly distributed below 8 km,
directly relevant to the lower tropospheric circulation, we follow the definitions and methodology
proposed by Tanaka et al. (2004) to obtain these tropical circulations at 850 hPa level for investigating
their contributions to the observed water vapor anomaly in the four El Niño events. The velocity potential
is divided as (Tanaka et al., 2004),

$$\Phi(x,y,t) = [\Phi(t,y)] + \overline{\Phi^*}(x,y) + \Phi^{*\prime}(x,y,t) \tag{7}$$

where $x$, $y$ and $t$ are the longitude, latitude and time, respectively. The square brackets and asterisk denote
the zonal mean and the deviation from the zonal mean, respectively, and the overbar and prime denote the
annual mean and the departure from the annual mean, respectively. The first term on the right of Eq. (7) is
the zonal mean component of the velocity potential field, defined as the Hadley circulation because this
circulation, driven by the large-scale meridional differential heating, may be treated as axisymmetric. The
second and third terms on the right are the annual mean of the deviation from the zonal mean and the
deviation from the annual mean, respectively. The third term is regarded to be the monsoon circulation
since the monsoon circulation has the conspicuous seasonal variability as the sea-land heat contrast
changes. The second term is referred to as the Walker circulation. The separation is not perfect for the
Walker circulation without seasonal variation, as pointed out by Tanaka et al. (2004). The Walker
circulation is induced by the different SST along the equator. Considering that the El Niño usually lasts for
more than a year with the maximum ONI in winter, we chose the period of June to the next May to
estimate the Walker circulation, and then obtain the Walker circulation anomaly during El Niño relative to
its climatic average. In this way, the problem may not be very serious. The definitions and decomposition
of the tropical circulations have extensively been used to study the influences of SST warming pattern on
the interannual variation and long-term trend of the Hadley, Walker and monsoon circulations in
association with hydrological cycle (Tanaka et al., 2005; Park and Sohn, 2008; Li and Feng, 2013; Ma and
Xie, 2013).
We firstly calculate the divergence field of the horizontal wind at 850 hPa from 2005 to 2019 by
using the reanalysis horizontal wind data, and then the velocity potential is deduced according to Eq. (6),
which is equivalent to solving Poisson equation. Next, according to Eq. (7), the velocity potential filed is
decomposed into the Hadley, Walker and monsoon circulation components. In this way, their monthly
climatic mean is derived from their time series, respectively. Figure 4 presents the climatic means of the
velocity potential and divergent wind fields in DJF. We choose the velocity potential as the proxy of the
circulation intensity, thus the intensity of the tropical circulation in winter can clearly be seen from Fig. 4.
The prominent negative peak of about $-81 \times 10^5$ m$^2$ s$^{-1}$ in the velocity potential is situated in the western
Pacific warm pool, thus there is the convergence center of horizontal wind field, which induces the rising
motion in the lower troposphere over the region, including the five radiosonde stations. Hence, the
atmospheric water vapor is abundant in this region due to the transport by the strong ascending flow. On
the contrary, the maximum velocity potential of $48 \times 10^5$ m$^2$ s$^{-1}$ appears in the northeast Pacific Ocean and
the southern part of the North American continent, meaning a downward motion associated with the
divergence center over there, as well as less water vapor relative to the western Pacific warm pool region.
**4.2 Atmospheric Circulation Anomalies**
Next, we focus on the tropical circulation anomaly in the four El Niño events. Figure 5 illustrates the
velocity potential and divergent wind anomalies at 850 hPa in the four winters. Here, we define the
velocity potential value as the circulation index with the units measured by $10^5$ m$^2$ s$^{-1}$, and accordingly,
the velocity potential anomaly is regarded as the index of the circulation anomaly. As a consequence, the
positive index of the circulation anomaly indicates the weakened convergence and rising motion or the
strengthened divergence and sinking motion, and vice versa for the negative index of the circulation
anomaly. Hence, the positive and negative indices mean the decrease and increase of water vapor in the
troposphere due to the vertical transport change, respectively. In Fig. 5, the positive index of the
circulation anomaly occurs in the western Pacific, especially in the 2009/10 and 2015/16 El Niño winters,
thus the ascending motion is suppressed over there, and the negative water vapor anomalies are recorded
in the radiosonde observation. On the contrary, there is the negative index in the equatorial eastern
Pacific, which causes that the descending flow is suppressed. Correspondingly, the positive CWV
anomaly over the equatorial eastern Pacific can be seen from Fig. 2.
According to Eq. (7), we calculate the velocity potential of the Hadley, Walker and monsoon
circulations and their anomaly indices at 850 hPa from the reanalysis data. Figure 6 presents the velocity
potential and anomaly index of the Hadley circulation in the four El Niño winters. Now that the Hadley
circulation is a tropical circulation driven by the meridional differential heating in the global radiative
process (Oort and Yienger, 1996), this large-scale circulation is very similar in different winters with the
circulation index increasing from the negative peak at about 12°S to positive peak at 23°N, and is little
affected by El Niño with the anomaly index less than $2\times10^5$ $m^2 s^{-1}$, or 2 units. Even so, the pattern of the
Hadley circulation anomaly is distinguished between the EP El Niño and CP El Niño. During the 2018/19
(2009/10) CP El Niño winters, the index of the Hadley circulation anomaly is positive over the entire
tropics with the maximum of 1.74 (1.65) units at 3°N (2°N). Whereas, in the 2006/07 and 2015/16 EP El
Niño winters, the positive index is located at about 5°N-30°N, and the negative index occurs over about
30°S-5°N. Li and Feng (2012) suggested that the different patterns of the Hadley circulation anomalies

between the CP and EP El Niños are associated with the contrasting underlying thermal structure changes because the maximum of the zonal-mean SST anomalies is moved northward to about 10°N in the CP event relative to the maximum around the equator in the EP event. At the five radiosonde sites, the averaged anomaly index is 0.29, 1.56, 0.65 and 1.37 units in the 2006/07, 2009/10, 2015/06 and 2018/19 winters, respectively, indicating that the Hadley circulation is too stable to have a significant impact on the water vapor variation.

Figure 7 depicts the velocity potential and anomaly index of the Walker circulation at 850 hPa in the El Niño winters. Relative to the Hadley circulation, the Walker circulation is the local circulation formed over the tropical Pacific with intense ascending flow in the western Pacific and descending flow in the eastern Pacific, thus the circulation has a high variability with the SST anomaly caused by ocean current. As the Walker circulation is directly related to ENSO, the scenario of the Walker circulation anomalies is roughly consistent with each other among the four El Niño events. In general, the positive and negative indices of the Walker circulation anomaly are located in the western and eastern Pacific, opposite to the circulation index, respectively, which illustrates that the Walker circulation anomaly in El Niño suppresses the strong rising in the western Pacific and sinking in the eastern Pacific. Nevertheless, the strength of the circulation anomaly is the significant difference among the four events. In the 2015/16 winter, the Walker circulation anomaly, with the peak indices as large as 26.8 and -27.7 units in the equatorial Pacific, are much stronger than in the other three winters. Hence, the Walker circulation variation plays a key role in the CWV anomaly during the 2015/16 super El Niño event.

The velocity potential and anomaly index of the monsoon circulation in the four El Niño winters are plotted in Fig. 8. The monsoon circulation in the lower atmosphere blows from the land to the sea in winter, thus it can be seen from Fig. 8 that the pattern of the monsoon circulation is evidently different

from that of the Walker circulation shown in Fig. 7. The anomaly of the monsoon circulation is sensitive

to the type of El Niño, which is also distinguished from that of the Walker circulation. Early studies

showed that the CP and EP El Niños have different effects on the Indian and eastern Asian monsoon

rainfall (Weng et al., 2008; Wang et al., 2013). The monsoon circulation anomaly in the radiosonde

stations has the index around zero in the EP El Niño events, which is far weaker relative to the large

positive index in the CP El Niño events, similar to previous investigation (Fan et al, 2017). In the

2009/10 El Niño event, the pronounced anomaly with the peak index of 17.8 units takes place in the

western Pacific, which implies that the monsoon circulation anomaly has an important influenced on the

negative water vapor anomaly in the radiosonde observation.

**4.3 Contribution to Water Vapor Anomaly**

We estimate the contributions of the Hadley, Walker and monsoon circulation anomalies to the

water vapor anomaly observed by the radiosonde in the four El Niño events by means of comparing the

indices of the circulation anomalies. Figure 9 illustrates the indices of the circulation anomalies at 850

hPa and the CWV anomalies derived from the radiosonde and reanalysis data, and these circulation

anomaly indices and CWV anomalies are the values averaged at the five radiosonde sites in winter. It can

be seen from Figure 9 that qualitatively, the CWV anomalies in the reanalysis and radiosonde data

increase with the increasing index of the total circulation anomaly. As discussed above, the contribution

of the Hadley circulation anomaly is very small with the maximum of only 1.56 units in the 2009/10

event. The anomaly of the Walker circulation makes a considerable contribution in each case, especially

for the EP El Niño events, it is the strongest in the three tropical circulation anomalies. The index of the

Walker circulation anomaly counts for 92.3% of the total anomaly index (23.89 units) in the 2015/16 El

Niño winter, and even exceeds the total index in the 2006/07 event owing to the negative anomaly of the

monsoon circulation. The anomaly of the monsoon circulation shows an evident change from one event
to another because it is sensitive to the local heat contrast and the El Niño shift. In the western Pacific,
the CP El Niño can lead to the obvious positive anomaly of the monsoon circulation. The index of the
monsoon circulation anomaly is about 69.7% (44.7%) of the total anomaly index in the 2009/10 (2018/19)
CP El Niño winter. Consequently, for the two intense El Niño events, the water vapor anomaly is caused
mainly by the Walker circulation anomaly in the 2015/16 EP event but by the monsoon circulation
anomaly in the 2009/10 CP event, respectively. The Walker and monsoon circulation anomalies nearly
equally (oppositely) contribute to the CWV anomaly in the 2018/19 (2006/07) event. Therefore, except
the Hadley circulation anomaly, the Walker and monsoon circulation anomalies may have the
considerable differences in the contributions to the water vapor variation in different El Niño events. In
addition, in the 2015/16 and 2018/19 winters, the reanalysis CWV anomalies of -4.34 and -1.30 kg m$^{-2}$
are roughly consistent with -4.46 and -1.54 kg m$^{-2}$ in the radiosonde observation, respectively. However,
in the first two events, there is a distinct difference of the CWV anomaly between the reanalysis and
radiosonde data, and we will discuss the discrepancy in detail below.
In order to obtain the general features of water vapor and circulation anomalies in the EP and CP El
Niño events, we extend the reanalysis data back to 1979 to examine two types of composite El Niño
events. There are six EP El Niño events in the winters of 1982/83, 1986/87, 1991/92, 1997/98, 2006/07
and 2015/16, and five CP El Niño events in the 1994/95, 2002/03, 2004/05, 2009/10 and 2018/19 winters
for 41 years from 1979 to 2019, which are averaged as the composite EP and CP El Niños, respectively.
We calculate the CWV anomalies in the two composite events based the climatic mean CWV in 41
winters, and the corresponding velocity potential and divergent wind anomalies of the Walker, monsoon
and total circulations from the reanalysis horizontal wind at 850 hPa, which are shown in Fig. 10. The
Hadley circulation anomaly (not presented) is very small, and its patterns in the composite EP and CP El
Niños are also analogous to those in the EP and CP events shown in Fig. 6, respectively. On the whole,
Fig. 10 illustrates that the total circulation anomaly is stronger in EP event than in CP event, and then the
CWV anomaly is larger in EP event relative to that in CP event. The Walker circulation plays an
important role in the total circulation anomaly, especially in EP El Niño. Despite significant variability
from one event to another, the monsoon circulation anomaly has not only a larger proportion of the total
anomaly but also slightly higher intensity in CP El Niño than in EP El Niño. At the five radiosonde
stations, the composite events indicate that the CWV anomaly is about -4.36 and -1.74 kg m$^{-2}$ in EP and
CP El Niños, respectively. The index of the Walker circulation anomaly accounts for about 75.8% (47.8%)
of the total anomaly index in EP (CP) El Niño, while for the monsoon circulation, the anomaly index of
6.16 (4.66) units contributes to 49.6% (18.4%) of the total anomaly index in CP (EP) El Niño. Therefore,
the relative importance of the Hadley, Walker and monsoon circulation anomalies in the composite El
Niños is roughly in accord with that in the case study above. In addition, at the radiosonde sites, the CP
El Niño can generally cause an intense monsoon circulation anomaly, which is comparable to and even
larger than the Walker circulation anomaly, thus the CP El Niño in the winter of 2009/10 may induce a
quite strong monsoon circulation anomaly now that the 2009/10 event is the strongest CP El Niño from
the 1980s, as observed by satellite (Lee and Mcphaden, 2010).

**5 Changes in Cloud and OLR**
Using the cloud occurrence from the CALIPSO during June 2006 to December 2016, we calculate
tropical cloud fraction between 0°N and 15°N in the 2006/07, 2009/10 and 2015/16 winters and its
climatic mean in winter, which is shown in Fig. 11. We also compute the OLR anomalies over 30°S-30°N
in the four El Niño winters based on the monthly OLR data between 2005 and 2019. Figure 12 shows the
OLR anomalies in the four El Niño events. In the western Pacific, the strong rising flow carries abundant
water vapor to high level due to the convergence of horizontal wind field in winter, as shown in Fig. 4,
and then the water vapor condenses to form clouds as it cools, thus there is clouds over the tropical
western Pacific. In the El Niño events, the cloud amount decreases from about 80°E to 160°E but tends to
increase between about 160°E to 120°W because of the tropical circulation changes. Owing to the
reflection effect of cloud on OLR, the OLR change is opposite to the variation of cloud amount. In the
2009/10 and 2015/16 strong El Niño winters, the OLR is obviously enhanced in the tropical northwest
Pacific and significantly reduced in the equatorial mid-eastern Pacific as the cloud occurrence changes.
Hence, the cloud and OLR have a clear response to the water vapor anomaly in the El Niño events.
As described above, the reanalysis CWV anomaly at the radiosonde stations in the 2009/10 winter
has an almost same intensity as that in the 2015/16 winter, but the radiosonde observation indicates that
the water vapor reduction is evidently less in the 2009/10 winter than in the 2015/16 winter. As shown in
Figs. 11 and 12, the satellite observation shows that there exist less cloud occurrence and more OLR at
the radiosonde stations in the 2015/16 winter compared with in the 2009/10 winter. Therefore, this
supports the radiosonde observation that the water vapor over the radiosonde stations in the 2009/10
winter may be moister than in the reanalysis.

**6 Discussion**
In the ERA5 reanalysis data, water vapor is calculated by a humidity analysis scheme introduced by
Hólm (2003), which involves nonlinear transformation of the humidity control variable to render the
humidity background errors nearly Gaussian. The transformation normalizes relative humidity
increments by a factor that varies as a function of background errors of relative humidity and vertical

level (Dee et al., 2011). For the ERA5 humidity analysis, measurements from radiosonde, surface synoptic observation, aircraft, and satellite observations are assimilated (Andersson et al., 2007). To date, the reliability and accuracy of ERA5 water vapor products have extensively been estimated. Overall, ERA5 retrieved precipitable water vapor (PWV) performs well over the Indian Ocean (Lees et al, 2020), central Asia (Jiang et al, 2019), Antarctic (Ye et al, 2007), East African tropical region (Ssenyunzi et al, 2020) and Varanasi (Kumar et al, 2020) via comparisons with ground-based observations, satellite retrievals and other reanalysis datasets. Nevertheless, some discrepancies can be noticed over small tropical islands characterized by steep orography (Lees et al, 2020), and it is reported that although PWV from the ERA5 reanalysis is in good agreement with the retrieval from Global Navigation Satellite System over 268 stations, a bias of 4 mm PWV in the southwest of South America and western China due to the limit of terrains and fewer observations (Wang et al, 2020).

Since the CWV anomalies look more or less different between the radiosonde and reanalysis data, we compare the CWV in the ERA5 reanalysis with that in the radiosonde and satellite observations at the five stations, and attempt to explain the different CWV anomalies between the reanalysis data and radiosonde observation in the 2006/07 and 2009/10 events. By using the reanalysis data and measurements of radiosonde and AIRS on Aqua satellite for the 15 year period from 2005 to 2019, we calculate the monthly mean CWV at the five radiosonde sites, and Fig. 13 depicts the monthly mean CWV in winter as scatterplots of the reanalysis vs. radiosonde data and the reanalysis vs. AIRS data. And then the climatic mean difference is derived from these monthly mean CWV series in 2005-2019, which is also presented in Fig. 13. At the five stations, the monthly mean CWV in winter is distributed between 30 and 60 kg m$^{-2}$ in all the three datasets, and the CWV is obviously shifted to the low values in the El Niño winter, indicating the negative anomaly in the El Niño event. The correlation of the mean CWV

series between the reanalysis and observations is quite high with the minimum coefficient of 0.88, and all
the root mean square (RMS) of the mean CWV differences between the reanalysis and observations is
less than 2.32 kg m$^{-2}$. Meanwhile, the difference of the climatic mean CWV is mainly concentrated in the
range of 0-2 kg m$^{-2}$ except several months at the Guam station, thus the relative difference of the monthly
mean CWV between the reanalysis and observations is generally smaller than 5%. These comparison and
analysis confirm a fine confidence level of the ERA5 reanalysis and observational datasets. Nevertheless,
there are still very small discrepancies among these data, and the discrepancy is relatively larger between
the radiosonde and reanalysis data than between the satellite and reanalysis data, which may be attributed
to a possible cause of different sampling times between the radiosonde and AIRS. It can be noted from
Fig. 13 that the red dots representing the reanalysis vs. radiosonde data in the 2009/10 winter show a
relatively scatter around the symmetric axis, indicating a relatively large discrepancy of the CWV
anomalies between the reanalysis data and radiosonde observation in this event, as previous reports of
some discrepancies over small tropical islands or in the region with fewer observations (Lees et al, 2020;
Wang et al, 2020). As comparison to the reanalysis data, the CWV derived from AIRS also shows the
largest difference of 1.31 kg m$^{-2}$ in the 2009/10 event, while the differences are less than 1 kg m$^{-2}$ in the
other three events.

Based on specific humidity in the reanalysis and radiosonde data, the CWV is calculated to be 44.87

(44.10), 43.06 (40.23), 41.16 (39.83) and 44.07 (42.87) kg m$^{-2}$ in the radiosonde (reanalysis) data in the
2006/07, 2009/10, 2015/16 and 2018/19 events, respectively. In fact, the relative difference of the CWV
between the radiosonde and reanalysis data is very small with only 1.7% in the 2006/07 winter, and 6.6%
in the 2009/10 winter. The CWV average in winter is 45.61 (44.17) kg m$^{-2}$ in the radiosonde (reanalysis)
data from 2005 to 2019, thus the CWV anomaly in the radiosonde (reanalysis) data is -0.74 (-0.07) kg
m$^{-2}$ in the 2006/07 event, and -2.55 (-3.94) kg m$^{-2}$ in the 2009/10 event. This causes that the discrepancy
of the CWV anomaly looks considerably large in Fig. 9, especially in the 2006/07 event, but the
differences of both the CWV and CWV anomaly values are small between the radiosonde and reanalysis.
Even so, the relatively large discrepancy between the reanalysis data and the radiosonde and AIRS
observations in the 2009/10 event, as shown in Figs. 1 and 13, and the cloud and OLR measurements in
Figs. 11 and 12 seem to suggest that the reanalysis data underestimates the tropospheric water vapor over
the radiosonde stations in the 2009/10 winter.

**7 Summary**
In the paper, we report the significantly negative water vapor anomaly in the troposphere during the
four El Niño winters at the five radiosonde stations in the tropical western Pacific based on the
radiosonde and reanalysis data for 15 years from 2005 to 2019, and study the relationship between the
water vapor anomaly and the El Niño index and the contribution of the different tropical circulation
anomalies to the observed water vapor anomaly in the El Niño events.
The radiosonde observation shows that the negative water vapor anomaly arises in the El Niño
winters, in particular, an extremely negative anomaly in the 2015/16 super El Niño event. The prominent
specific humidity anomaly is concentrated below 8 km of the troposphere with the peak at the height of
about 2.5-3.5 km. The local CWV anomaly has a large negative correlation coefficient of -0.63 with the
ONI in the Niño3.4 region, but with a lag of about 2-3 months. The reanalysis data reveals that the
negative water vapor anomaly widely occurs in the tropical northwest Pacific, while correspondingly, the
positive anomaly takes place in the equatorial mid-eastern Pacific. The 2015/16 El Niño event, with
ONI=2.6, is the strongest during the 15 years, leading to the extreme anomaly in the water vapor over the
tropical Pacific.
The atmospheric water vapor from tropical sea water evaporation is affected not only by the SST, but
also by the vertical motion of the atmosphere which can transport the water vapor from the near-sea
surface to the high level. By using the definitions and method introduced by Tanaka et al. (2004), we
decompose the tropical circulation into the Hadley, Walker and monsoon circulations to estimate their
contributions to the observed water vapor anomaly in the four El Niño events. In general, the tropical
circulation anomaly in the El Niño winter is characterized by divergence (convergence) at 850 hPa in the
tropical western (eastern) Pacific, thus the CWV decreases over the tropical western Pacific as the
ascending flow is suppressed. As the large-scale meridional circulation driven by the differential heating,
the variation of the Hadley circulation is pretty small with the anomaly index less than 2 units. At the
radiosonde stations, the anomaly of the Walker circulation makes a considerable contribution to the total
anomaly in all the El Niño winters, especially in the 2006/07 and 2015/16 EP El Niño event. The
monsoon circulation exhibits an obvious variability from one event to another, and its anomaly is large in
the 2009/10 and 2018/19 CP El Niño winters and small in the 2006/07 and 2015/16 EP El Niño winters.
Therefore, the observed water vapor anomaly is caused mainly by the Walker circulation anomaly in the
2015/16 super EP event but by the monsoon circulation anomaly in the 2009/10 strong CP event,
respectively. Based on the reanalysis data back to 1979, we examine the general features of water vapor
and circulation anomalies in the two types of composite El Niño events. The roles of the Hadley, Walker
and monsoon circulations in the composite EP and CP El Niños are consistent with those in the EP and
CP case events.
Because of the reduction in the upward transport of water vapor over the tropical western Pacific in
the El Niño events, the satellite observation shows that relative to the climatic means, the cloud decreases,
and the OLR is accordingly strengthened, in particular, during the strong El Niño winters of 2009/10 and
2015/16. In addition, a detailed comparison of water vapor in the reanalysis, radiosonde and satellite data
shows a high confidence level of these datasets, nevertheless, the reanalysis seems to slightly
underestimate the water vapor over the five radiosonde stations in the 2009/10 winter.


**Data availability.** The radiosonde observation is provided by the NOAA at the website of
ftp://ftp.ncdc.noaa.gov/pub/data/ua/rrs-data/. The ERA5 reanalysis data is from the ECMWF at
https://cds.climate.copernicus.eu/cdsapp#!/home/. The Niño3.4 index is from the NOAA at
https://catalog.data.gov/dataset/climate-prediction-center-cpcoceanic-nino-index/. The OLR data is from
the NOAA at https://www.esrl.noaa.gov/psd/data/gridded/data.interp_OLR.html/. The cloud occurrence
monthly data is from the NASA at https://eosweb.larc.nasa.gov/project/calipso/cloud_occurrence_table/,
and the AIRS water vapor data is available from the NASA at http://disc.sci.gsfc.nasa.gov.

**Author contributions.** KH and MD proposed the scientific ideas. MD and KH completed the analysis and
the manuscript. SZ, CH, YG and FY discussed the results in the manuscript.

**Competing interests.** The authors declare that they have no conflict of interest.

**Acknowledgments.** This work was supported by the National Natural Science Foundation of China
(through grants 41974176 and 41674151).

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

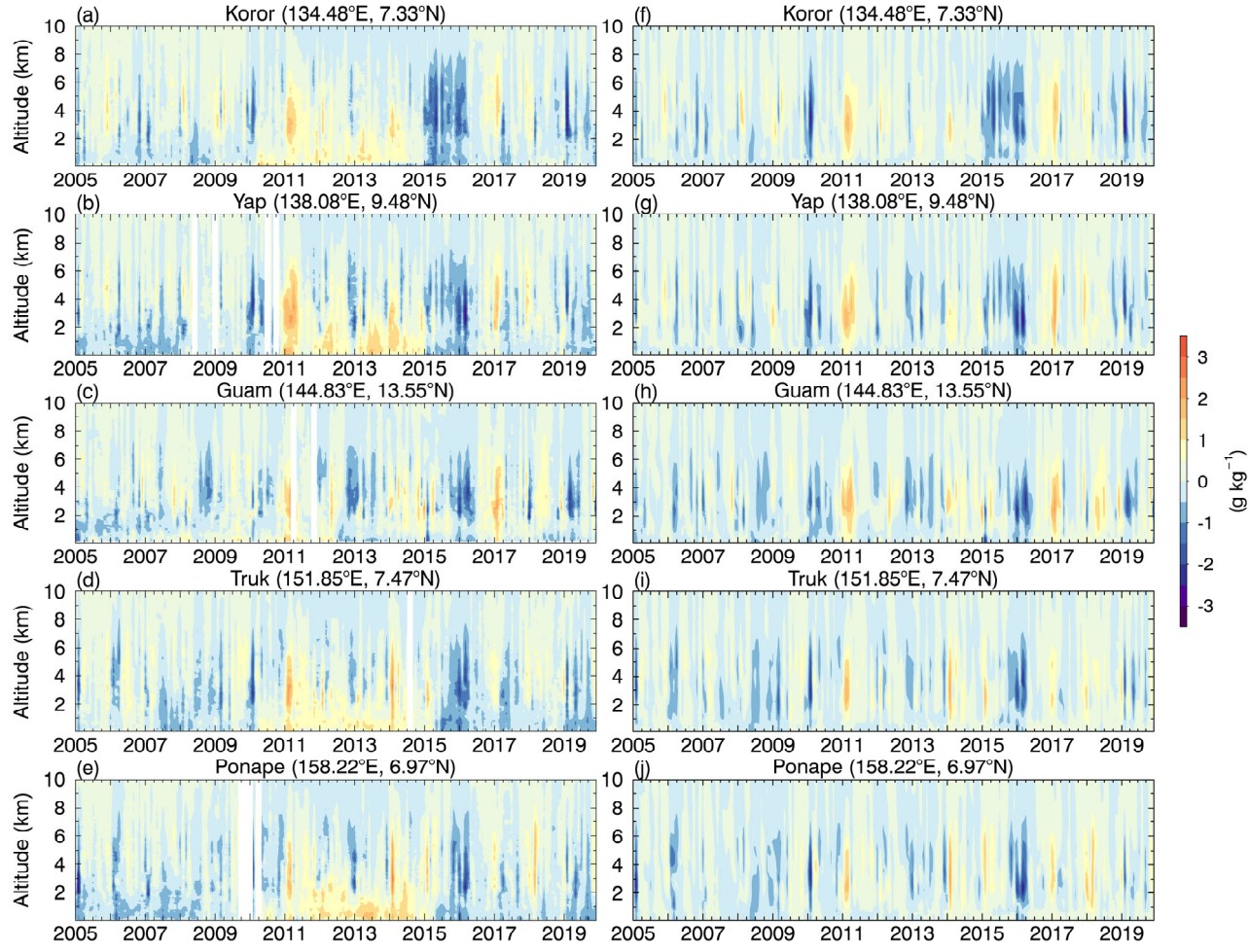

**Figure 1.** Specific humidity anomaly between January 2005 and December 2019 derived from (left) radiosonde observations and (right) ERA5 reanalysis data at (a, f) Koror, (b, g) Yap, (c, h) Guam, (d, i) Truk and (e, j) Ponape.

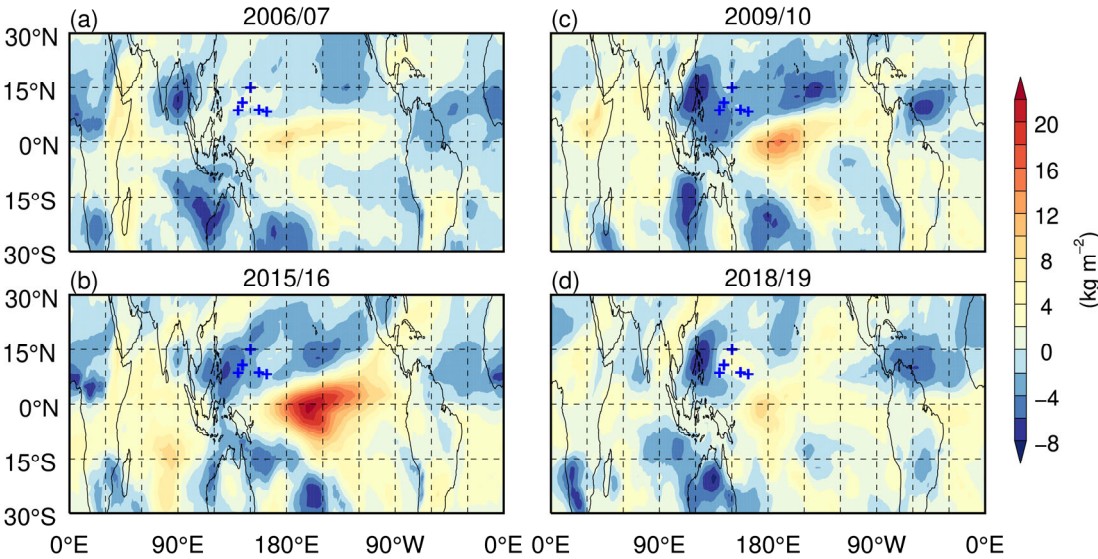


**Figure 2.** CWV anomalies averaged in (a) 2006/07, (b) 2015/16, (c) 2009/10 and (d) 2018/19 winters
derived from ERA5 reanalysis data. The blue plus denotes the five radiosonde stations. The four El Niño
events are classified into (left) EP El Niño and (right) CP El Niño.

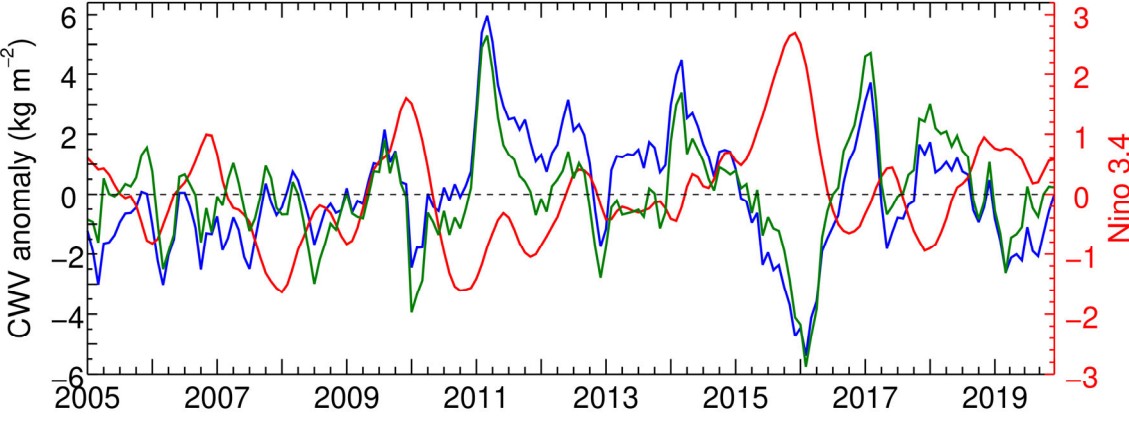

**Figure 3.** Time series of (red) ONI index and monthly mean CWV anomalies derived from (blue) radiosonde observation and (green) reanalysis data at five radiosonde stations.

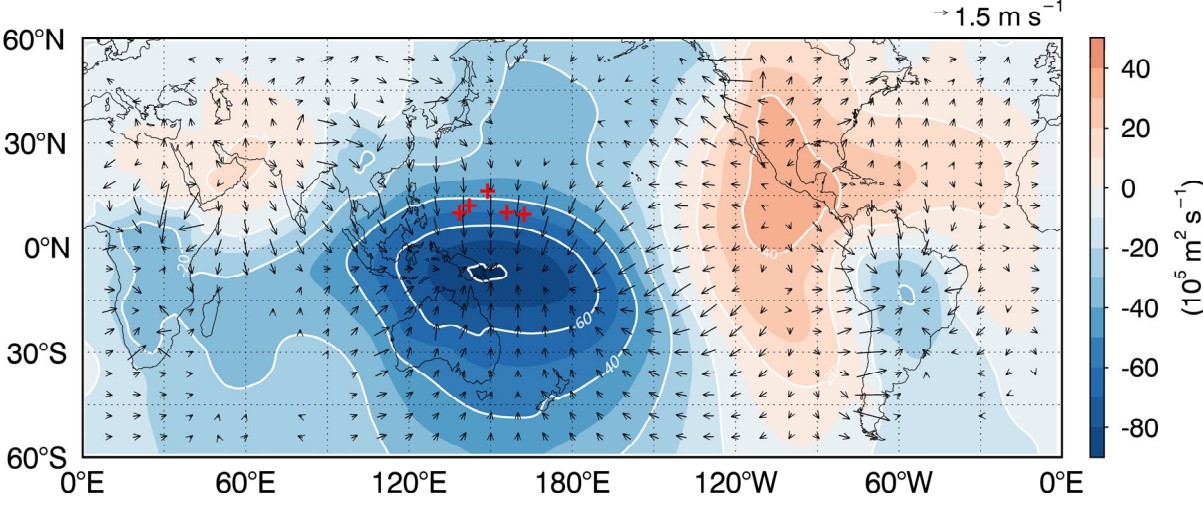


**Figure 4.** Climatic means of (shading) velocity potential and (arrow) divergent wind fields at 850 hPa in

DJF derived from reanalysis data during 2005-2019. The red plus denotes the five radiosonde stations.

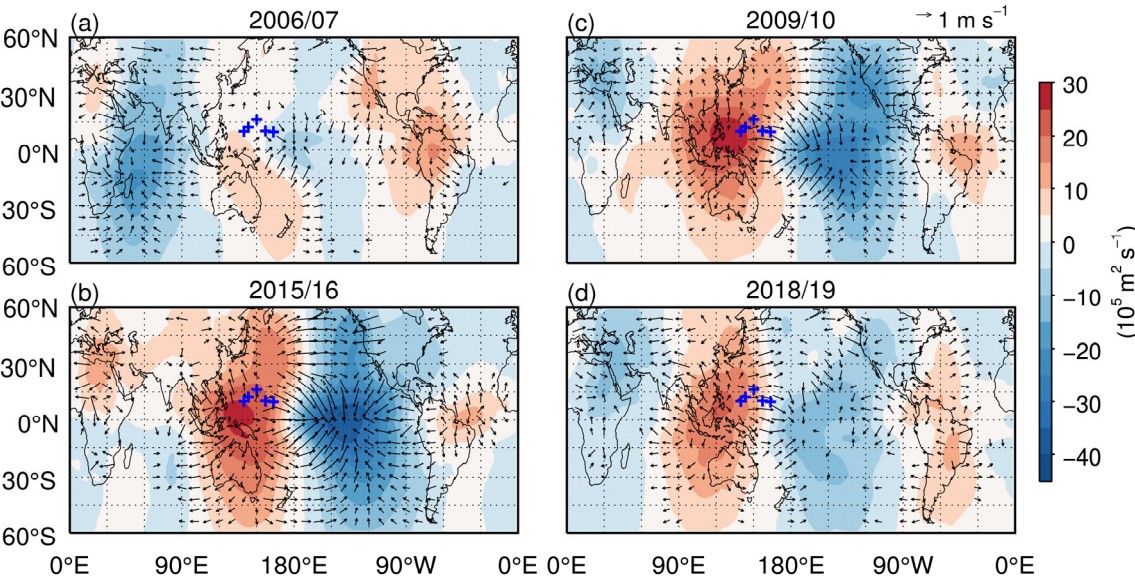


**Figure 5.** Anomalies of (shading) velocity potential and (arrow) divergent wind at 850 hPa in winters of (a)

2006/07, (b) 2015/16, (c) 2009/10 and (d) 2018/19. The blue plus denotes the five radiosonde stations.

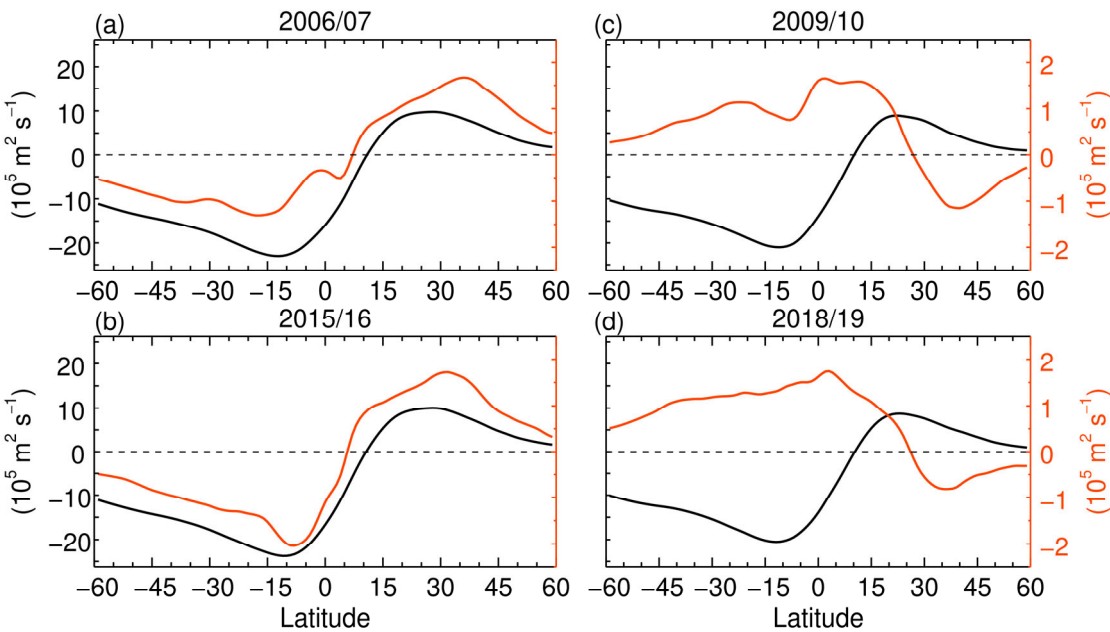


**Figure 6.** (Black)Velocity potential and (orange) anomaly index of Hadley circulation at 850 hPa derived

from reanalysis data in (a) 2006/07, (b) 2015/16, (c) 2009/10 and (d) 2018/19 winters.

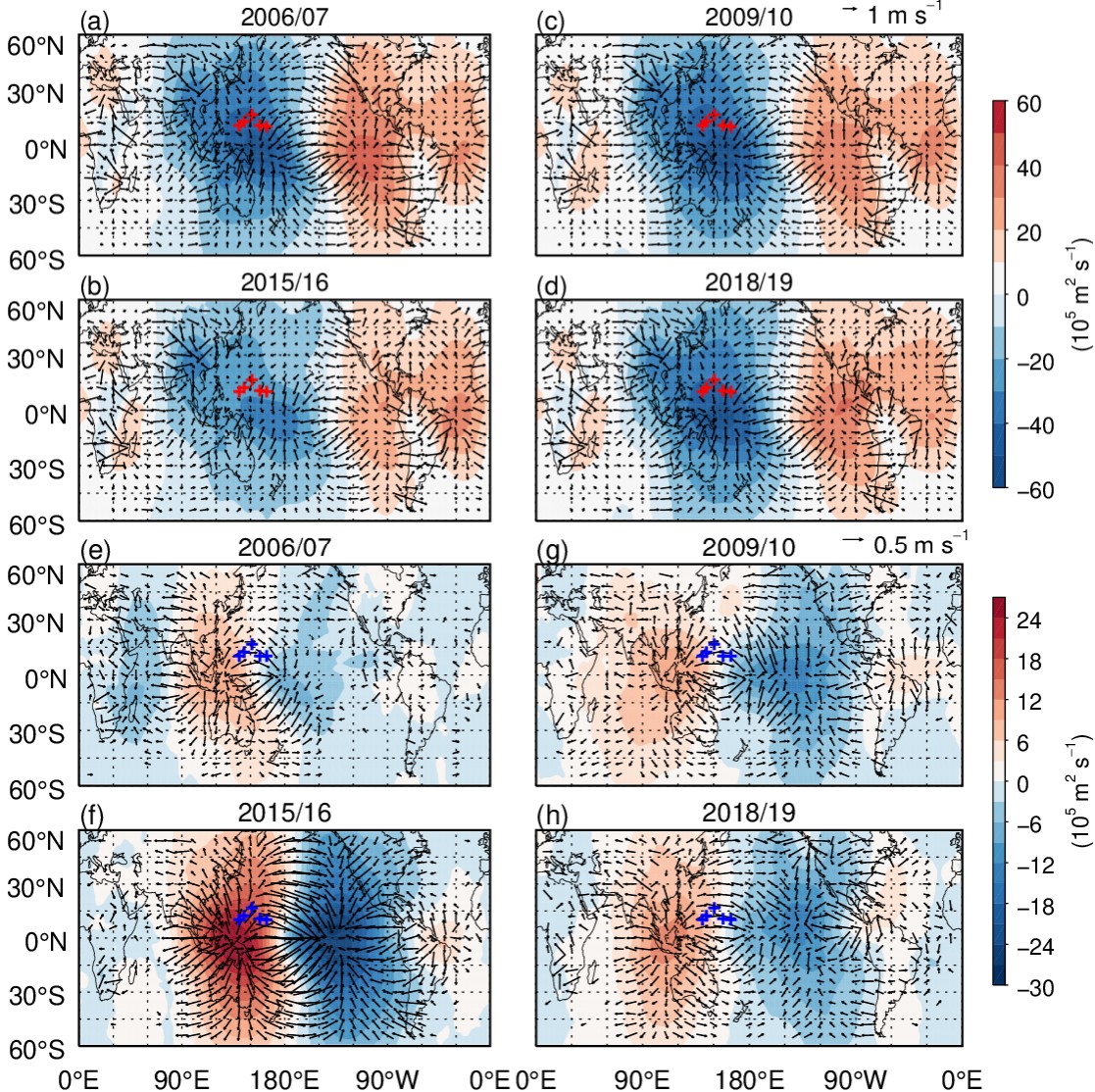


**Figure 7.** (shading) Velocity potential and (arrow) divergent wind of Walker circulation and their

anomalies at 850 hPa in (a, e) 2006/07, (b, f) 2015/16, (c, g) 2009/10 and (d, h) 2018/19 winters. Figure 7

(a-d) denotes the velocity potential and divergent wind, and Figure 7 (e-h) denotes their anomalies. The

red and blue plus denotes the five radiosonde stations.

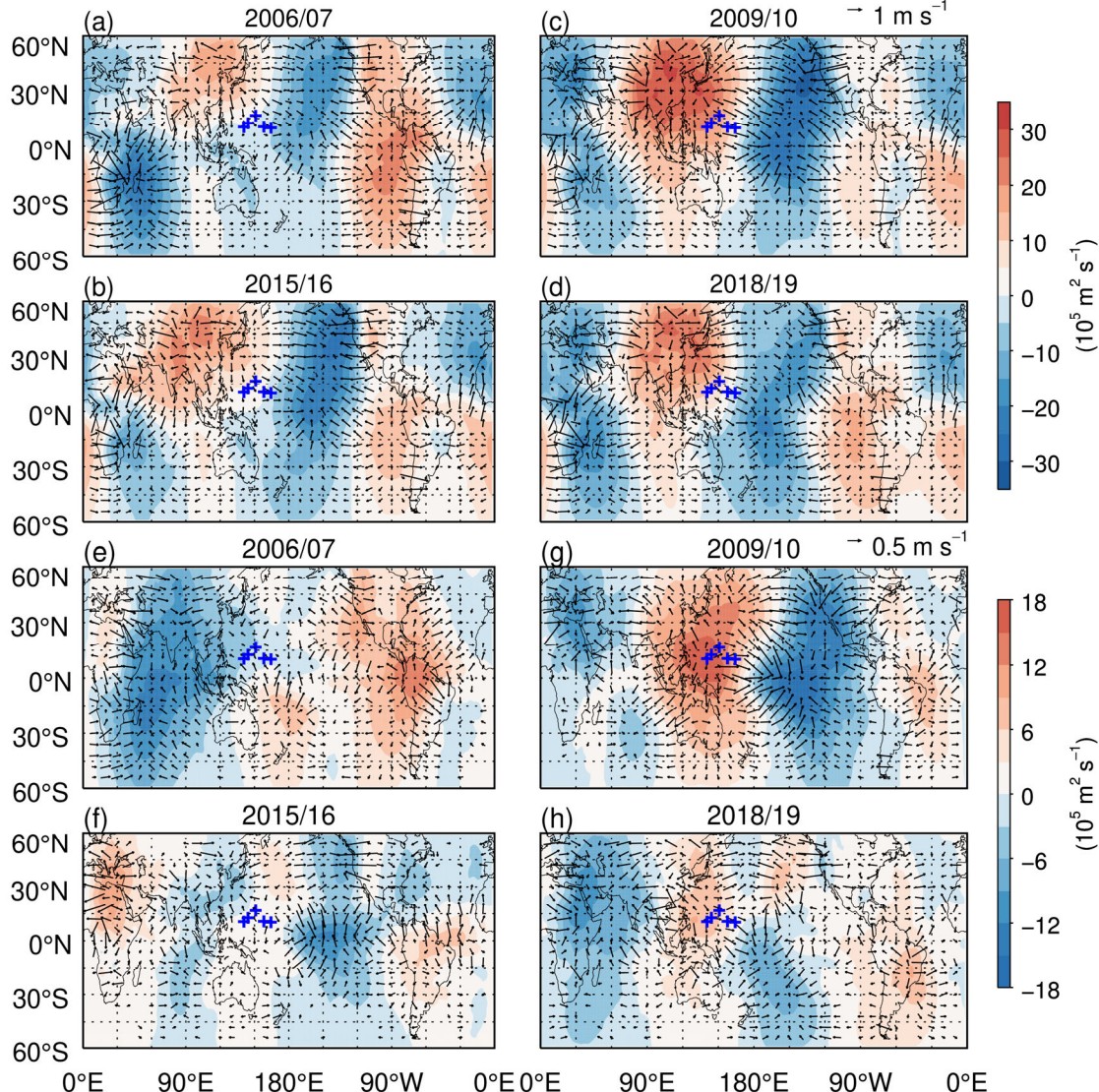

785

**Figure 8.** (shading) Velocity potential and (arrow) divergent wind of monsoon circulation and their

anomalies at 850 hPa in (a, e) 2006/07, (b, f) 2015/16, (c, g) 2009/10 and (d, h) 2018/19 winters. Figure

8(a-d) denotes the velocity potential and divergent wind, and Figure 8(e-h) denotes their anomalies. The

blue plus denotes the five radiosonde stations.

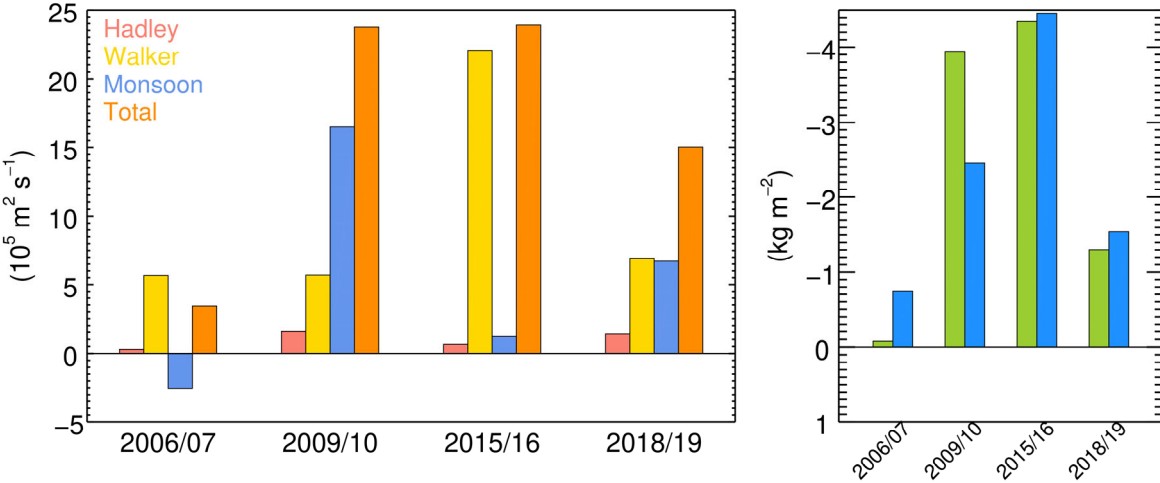

790

**Figure 9.** (Left) Indices of (red) Hadley, (yellow) Walker, (blue) monsoon and (orange) total circulation

anomalies and (right) CWV anomalies derived from (azure) radiosonde and (green) reanalysis data at five

radiosonde stations in four El Niño winters.

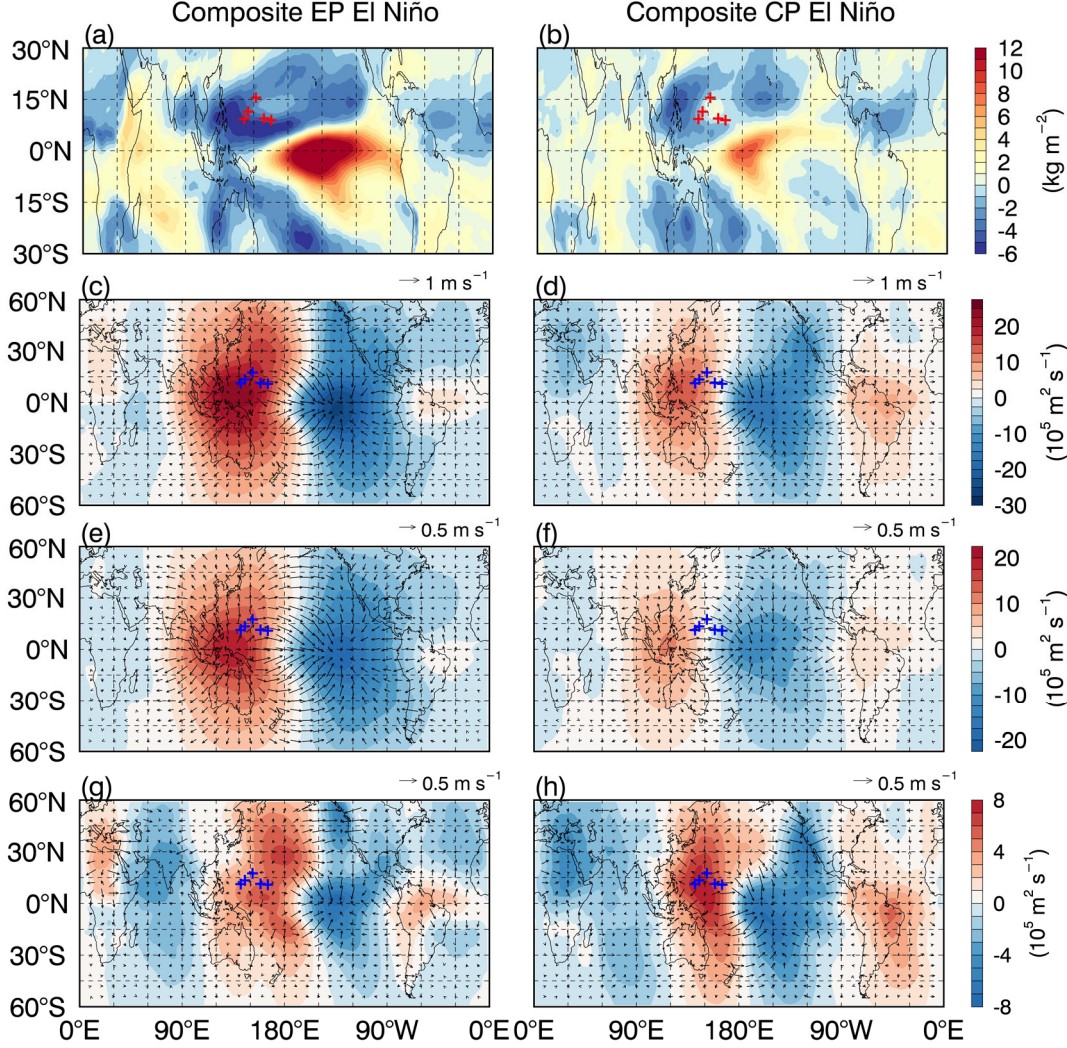

794

**Figure 10.** Anomalies of (a, b) CWV and velocity potential and divergent wind at 850 hPa in (c, d) total,

(e, f) Walker and (g, h) monsoon circulations for composite EP and CP El Niños derived from reanalysis

data. The left and right columns correspond to the composite EP and CP El Niños, respectively. The

shading and arrow in Fig. 10 (c-h) denote the velocity potential and divergent wind anomalies,

respectively. The red and blue plus denotes the five radiosonde stations.

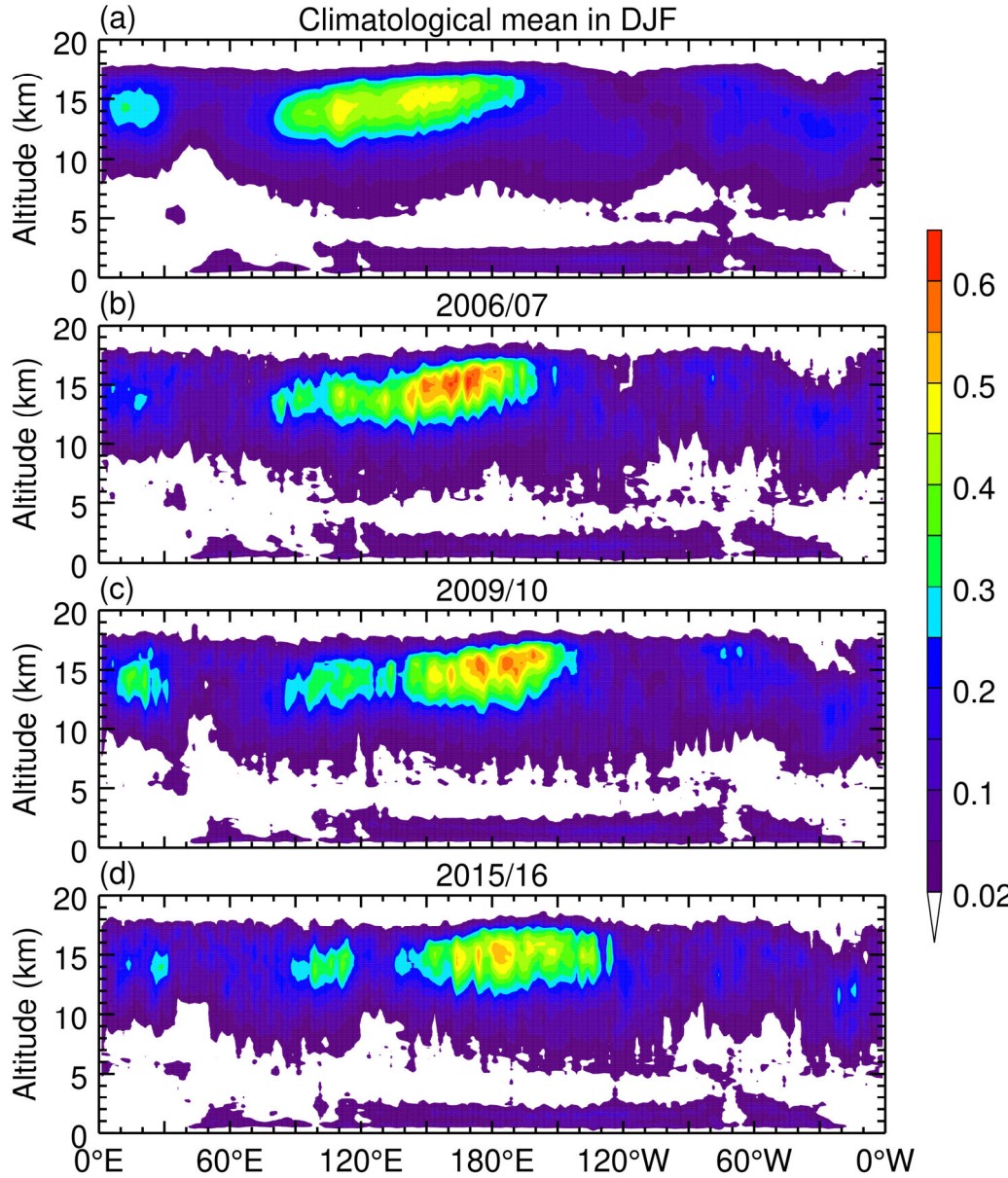

**Figure 11.** Distribution of cloud occurrence between 0°N and 15°N in (a) all winters, and (b) 2006/07, (c)

2009/10 and (d) 2015/16 winters derived from CALIPSO during June 2006 to December 2016.

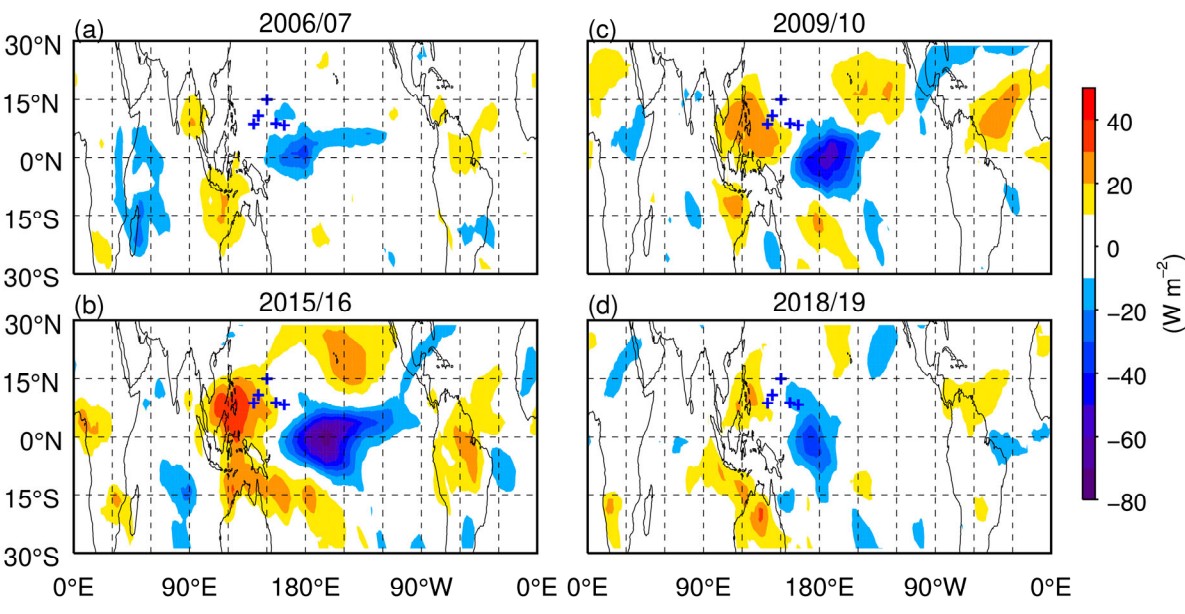


**Figure 12.** OLR anomalies averaged in (a) 2006/07, (b) 2015/16, (c) 2009/10 and (d) 2018/19 winters.
The blue plus denotes the five radiosonde stations.

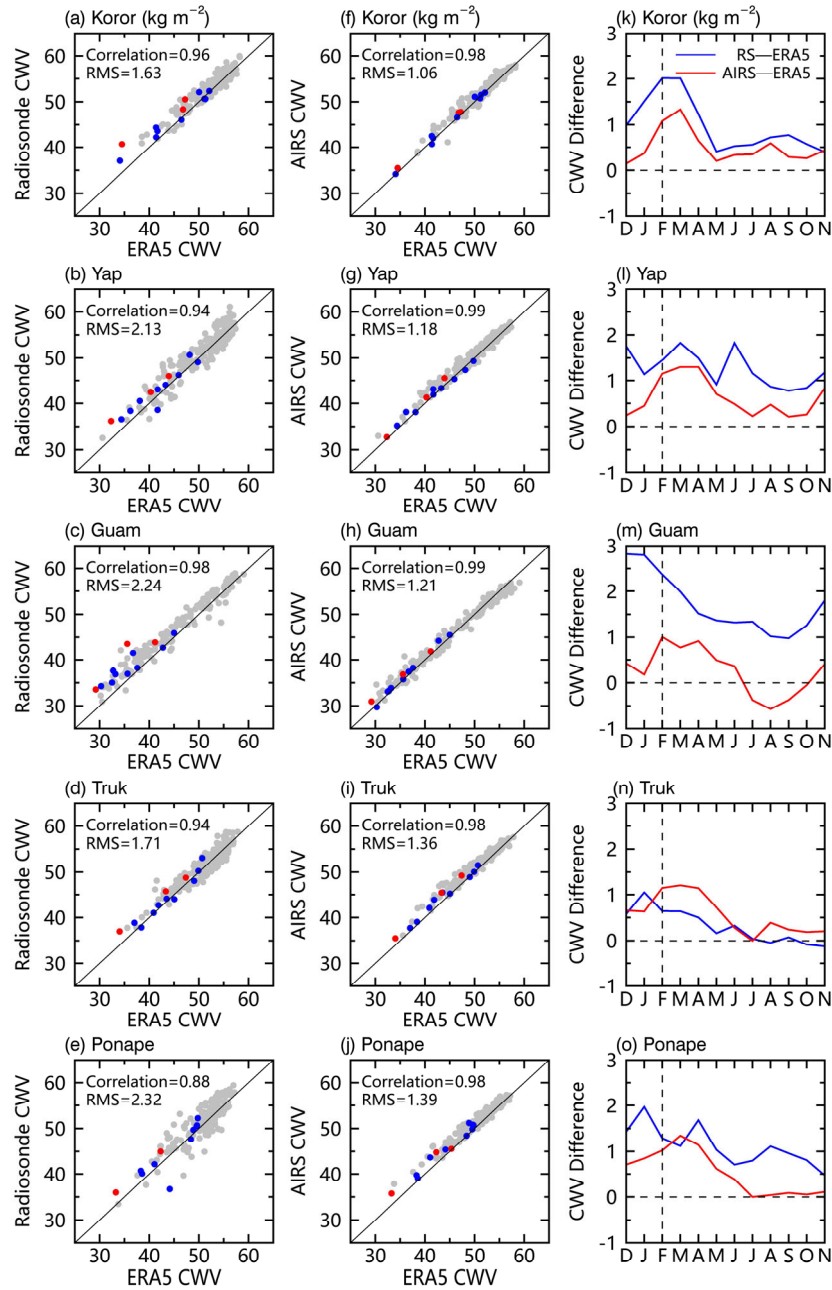


**Figure 13.** Scatterplots of monthly mean CWV in winter derived from (a-e) radiosonde and (f-j) AIRS

observations against corresponding CWV from ERA5 reanalysis and (k-o) climatic mean CWV difference

(blue lines) between radiosonde and ERA5 reanalysis data and (red lines) between AIRS and ERA5

reanalysis data at five stations during 2005-2019. In Fig. (a-j), the red, blue and gray dots denote the CWV

values in the 2009/10 winter, the 2006/2007, 2015/2016 and 2018/2019 winters, and the other winters,

respectively.