# Peer review of "Water vapor anomaly over the tropical western Pacific in El Niño winters from radiosonde and satellite observations and ERA5 reanalysis data"

_Atmospheric Chemistry and Physics, 2021_

## Author Comment (AC1)

**·Response to reviewer 1:**

We are deeply grateful to you for your valuable suggestions on our manuscript, which significantly improves the manuscript.

We have read these comments and suggestions seriously. According to your suggestions, we revised our manuscript, and presented a point-to-point reply to your comments. The comments from the referee are highlighted in blue. Our responses and a brief summary of related changes to the manuscript are given below. Meanwhile, we also provided a revision with tracking the changes.

**General comments**

In "Water vapor anomaly over the tropical western Pacific in El Niño winters from radiosonde and satellite observations" the authors use radiosonde observations from six sites in the Pacific as well as water vapor from the ECMWF reanalysis to show a correlation between column integrated water vapor and ENSO. They then use the velocity potential to determine the relative impacts of the Hadley, Walker, and monsoon circulations on 4 El Niño events (2006/07, 2009/10, 2015/16, and 2018/19). They find relatively little contribution to water vapor anomalies from the Hadley circulation, but find that variations in the Walker circulation drive the water vapor anomaly during these El Niño events, particularly during the 2015/16 event. Likewise, changes in the monsoon circulation were important for driving the water vapor anomaly for the 2009/10 event. While the importance of the Walker Circulation in driving ENSO-related changes in water vapor is well-known, this work does advance understanding of the relative impact of the different circulations on individual ENSO events and could be published in ACP once the deficiencies outlined below are addressed. Finally, while the authors' intent is, for the most part, clear, the manuscript needs to be closely read for grammar, as there are frequent minor errors that make understanding more difficult.

We thank you for your encouragement, and do our best to improve our manuscript according to the review and editor valuable suggestion.

**Major Concerns:**

1. A more thorough evaluation of the water vapor product from the reanalysis dataset is needed to support the results of this work. The only comparison between the reanalysis water vapor and observations shows major

discrepancies that the authors inadequately explain. Since the reanalysis water vapor underpins many of the conclusions in the paper, its accuracy is important and any uncertainties should be properly outlined. In addition to a more thorough comparison to the radiosonde observations discussed here (an equivalent figure to Figure 1 with the ECMWF data would make sense), comparison to satellite observations, preferably those independent of that assimilated for the reanalysis product, over the tropics is warranted. Are anomalies apparent in the ECMWF data evident in the satellite product? At the very least, the authors should cite previous work that analyzes the accuracy of the reanalysis product and discuss how biases/errors in the reanalysis could affect their results.

As you said, the reanalysis water vapor underpins many of the conclusions in the paper, thus the comparison and discussion of the reanalysis product is of significance.

According to your suggestion, we compare the water vapor in the reanalysis with that in the radiosonde and satellite observations and cite previous work to estimate the reanalysis product.

In the revised manuscript, the description of the satellite product is added in section 2 as follows,

"To assess the atmospheric water vapor as compare to the reanalysis data and the radiosonde observations, a further evaluation is carried out using Aqua atmospheric infrared sounder (AIRS) water vapor mass mixing ratio data from 2005-2019. AIRS is a hyperspectral infrared spectrometer orbiting on the national aeronautics and space administration (NASA) Aqua spacecraft launched in May 2002, which can provide accurate measurements of temperature, moisture, and other atmospheric variables (Aumann et al., 2003). The data used here is water vapor vertical profiles from Level 3 monthly standard gridded retrieval product version 6, AIRS3STM (Susskind et al., 2014), which is available at http://disc.sci.gsfc.nasa.gov. The water vapor data contains 8 levels from 1000 and 300 hPa with a latitudinal and longitudinal grid of 1°×1°, derived from the average of twice observations in two orbital overpasses per day. The ascending and descending orbits have equatorial crossing time at 13:30 local time (LT) and 1:30 LT, respectively."

W calculate the monthly mean specific humidity anomaly from the reanalysis data at the five stations, which is shown in Fig. 1 in the revised manuscript along with the radiosonde results. During the four El Niño winters, the reanalysis water vapor exhibits negative anomalies in the lower and middle troposphere, especially in the 2015/16 super event, which is consistent to the radiosonde observations. The discrepancy of water vapor between radiosonde and reanalysis data in 2009/10 winter also can be

seen in Fig. 1. The reanalysis specific humidity is slightly drier than the radiosonde observations, in particular, in Koror and Guam.

In the revised manuscript, we add the corresponding description for Fig.1 in Section 3.1 as follows, "We derive the monthly mean specific humidity anomaly from the reanalysis data at the radiosonde stations during the same period, which is also presented in Fig. 1. The ERA5 reanalysis shows water vapor anomaly scenario similar to the radiosonde observation. The negative anomalies in the four El Niño winters are obvious in the reanalysis data, especially the strong anomaly in the 2015/16 event."

A quantitative comparison of the reanalysis and observation will be presented in Section 6 of discussion.

[Figure]

**Figure 1.** Specific humidity anomaly between January 2005 and December 2019 derived from (left) radiosonde observations and (right) ERA5 reanalysis data at (a, f) Koror, (b, g) Yap, (c, h) Guam, (d, i) Truk and (e, j) Ponape.

And we add the discussion of the water vapor data quality as Section 6, which are presented as follows,

**"6 discussion**

[revised manuscript text omitted]

2. The authors also do not do an adequate job of justifying their method for determining the relative effects of the Hadley, Walker, and monsoon circulations based on the potential velocity. The method used here, based on the work of Tanaka et al. (2004), was designed based on upper tropospheric (~200 hPa) values of the potential velocity. All the previous work the authors cite (Tanaka, 2005; Park and Sohn, 2008; Ma and Xie, 2013) as examples of this method also use upper tropospheric velocity potential for this calculation. Here, however, the authors use the velocity potential at 850 hPa. While this could ultimately be fine, some justification needs to be given as to why the method is applicable in the lower free troposphere. Further, there should be more discussion as to why fields derived from the upper tropospheric potential velocity are less relevant for this work than those at 850 hPa.

As you pointed out, the method of the tropical circulation separation proposed by Tanaka et al. (2004) was designed for the upper troposphere (~200 hPa). Subsequently, Takemoto and Tanaka (2007) used this mothed to analyze the tropical Hadley, Walker, and monsoon circulations at 850 hPa of the lower troposphere, and compared the three circulations in the upper (~200 hPa) and lower troposphere (~850 hPa). Their results showed that although intensities of the circulations at the lower level are less than 50% of the upper troposphere, the ratios of these tropical circulations are comparable to those in the upper troposphere. The variational trends of the three circulation components in the lower troposphere are in agreement with those in the upper troposphere. In the vertical structure of the atmospheric circulations, the center of the convergent (divergent) in the lower level coincides with the center of the divergent (convergent) in the upper level. Hence, they suggested that the velocity potential intensities could be an index of each circulation in the lower troposphere without a notable influence from the surface.

We chose the potential velocity at 850 hPa rather than at 200 hPa, which is because atmospheric water vapor is mainly concentrated in the lower and middle troposphere, thus has a more directly relevant to the lower tropospheric circulation than to the upper tropospheric one. For example, when the horizontal winds are convergent in the lower atmosphere, the induced ascending flow carries out water vapor from the sea surface to high level to increase water vapor in the atmosphere, thus the physical scenario is clear. However, correspondingly, the wind field is divergent wind in the upper troposphere since the divergent winds in the upper troposphere are opposite direction to those in the lower troposphere, and then the induced vertical wind is upward only in the upper troposphere, which

cannot be directly associated with the water vapor change in the lower and middle troposphere.

According to your suggestion, we rewrite and add the sentences of "Based on the different driving mechanisms and movement features, Tanaka et al. (2004) introduced the definitions of the Hadley, Walker and monsoon circulations, which have an advantage to quantitatively evaluate the intensity of the three tropical circulations by means of the separation of the velocity potential into three orthogonal spatial patterns. Thus, we follow the definitions and methodology proposed by Tanaka et al. (2004) to obtain these tropical circulations for investigating their contributions to the observed water vapor anomaly in the four El Niño events." to be

"Based on the different driving mechanisms and movement features, Tanaka et al. (2004) decomposed the tropical circulation in the upper troposphere (200 hPa) into the Hadley, Walker and monsoon circulations, which have an advantage to quantitatively evaluate the intensity of the three tropical circulations by means of the separation of the velocity potential into three orthogonal spatial patterns. Subsequently, Takemoto and Tanaka (2007) used these circulation definitions to analyze the Hadley, Walker, and monsoon circulations at 850 hPa of the lower troposphere, and compared the three circulation components with those in the upper troposphere (200 hPa), which indicated that the velocity potential intensities could be an index of each circulation in the lower troposphere without a notable influence from the surface. Considering that atmospheric water vapor is mainly distributed below 8 km, directly relevant to the lower tropospheric circulation, we follow the definitions and methodology proposed by Tanaka et al. (2004) to obtain these tropical circulations at 850 hPa level for investigating their contributions to the observed water vapor anomaly in the four El Niño events."

**Minor Comments:**

1. Title: The title is somewhat misleading as satellite observations play an extremely minor role in the analysis, unless you are considering the water vapor assimilated into the ECWMF reanalysis. I would recommend changing the title to more accurately reflect the bulk of the work in the paper.

In the revised manuscript, the title is changed as "Water vapor anomaly over the tropical western Pacific in El Niño winters from radiosonde and satellite observations and ERA5 reanalysis data" in the revision, because 1) the satellite observation is used to estimate the reanalysis water vapor in detail, 2) Figure 13 associated with satellite data is added, and 3) the reanalysis is as an important data

in the study.

2. Line 51: Instead of America, it's probably better to say USA if you're referring to the country, or be more specific (e.g. North, South, Central America) if you are referring to the continent/region.

"America" is replaced as "continental USA" in the text.

3. Line 112: What percentage of observations were deemed to be outliers?

By following the analysis by Lanzante (1996), the outliers of temperature, wind and relative humidity are found to be about 0.09%, 0.08% and 0.02% of all observational data at the five stations in 15 years, respectively.

Thus, we add the corresponding description in the revised manuscript as follows,

"The outlier data are very few, and the outliers of temperature, wind and relative humidity account for only 0.09%, 0.08% and 0.02% of all observational data at the five stations during 15 years, respectively."

4. Line 116: I don't understand what you mean by "… and they are almost entirely from the several gaps of observations." Please reword this sentence.

"… and they are almost entirely from the several gaps of observations." is written as

"… and they are almost entirely from the several continuous observational missing rather than balloon burst below 10 km."

5. Line 129: Since water vapor plays such a major role in this paper more discussion needs to be included about how it is determined in the reanalysis. Is it part of the data assimilation scheme? If so, what satellite products are used for the water vapor assimilation? Has the water vapor product been evaluated?

In the ERA5 reanalysis product, atmospheric water vapor is an important part of the data assimilation scheme. As our response to Major Comments 1, in the revised manuscript, we add a description of the ERA5 water vapor product, cite previous studies to evaluate the accuracy of the product, and compare in detail the water vapor in the reanalysis with that in the radiosonde and satellite observations in Section 6 of discussion.

6. Line 148: You should list the satellite overpass time for NOAA18 and CALIPSO since these are both polar orbiting satellites.

> According to your suggestion, we add the depiction as
>
> "The OLR data is measured by the NOAA-18 satellite, which travel in sun-synchronous orbit with a 13:55 LT equatorial crossing time (Kramer, 2002)" for NOAA-18 satellite,
>
> and "The satellite has a sun-synchronous orbit with an equatorial crossing time around 1:30/13:30 LT (Stephens et al., 2002)" for CALIPSO.

7. Line 168: The phrase "… the observed water vapor also exhibit negative throughout the lower troposphere" does not make sense and needs to be reworded.

> The phrase is reworded as "…the observed water vapor also exhibits the negative anomalies in the lower and middle troposphere".

8. Line 182: How well does the CWV for the radiosonde data compare to that from the reanalysis? As discussed above, a more thorough analysis than that shown in Figure 9 is warranted.

> According to your suggestion, we conduct a thorough comparison of the CWV among the reanalysis data and the radiosonde and satellite observations, as presented in our response to Major Comments 1. As shown in the added discussion in Section 6, the CWV is ~ 40-50 kg m$^{-2}$ in both the radiosonde and reanalysis data due to the abundant water vapor over the tropical Pacific, and the discrepancy among the reanalysis, radiosonde and satellite data is very small, which confirms a fine confidence level of the ERA5 reanalysis and observational datasets.
>
> On the whole, the CWV anomaly of about 0-4 kg m$^{-2}$ is small relative to the CWV. Some discrepancies between the reanalysis and observations over small tropical islands or in the region with fewer observations are also reported in previous studies (Lees et al, 2020; Wang et al, 2020).

9. Line 197: The phrase "… but tends to vary in opposite to the ONI" needs to be reworded.

> The phrase is addressed as "…but a negative correlation to the ONI".

10. Line 227: As described above, this explanation is insufficient to justify using 850 hPa to characterize the

circulation as previous methods all use the upper troposphere for this type of analysis. Please explain this choice more thoroughly and demonstrate that this method is applicable to the lower as well as upper troposphere.

Besides the explanation in response to Major Comment 2, we rewrite the sentence of "Thus we selected the velocity potential at 850 hPa to represent the characteristics of the tropical circulations in the lower troposphere." to be,

"Because atmospheric water vapor comes mainly from the lower atmosphere through transport of ascending flow, we selected the velocity potential at 850 hPa to represent the characteristics of the tropical circulations in the lower troposphere since the pressure level was extensively used to investigate the lower atmospheric circulation (Wang, 2002; Weng et al., 2008; Zhao et al., 2010)."

11. Line 262: Do you mean climatic "mean"?

Yes, in the revision, "their monthly climatic normal" is changed as "their monthly climatic mean".

12. Line 295: Can you explain/hypothesize as to what is causing the difference in the Hadley circulation anomaly between the CP and EP El Niños?

Based on the study by Li and Feng (2012), the patterns of the zonal-mean SST anomalies are different between the CP and EP El Niños. In the EP El Niño, the zonal-mean SST anomalies is symmetric with respect to the equator with a maximum around the equator. However, the zonal-mean SST anomalies associated in the CP event shows an asymmetric structure with a maximum around about 10°N. Hence, the contrasting underlying thermal structures in the EP and CP events have different impacts on the Hadley circulation and its anomaly.

In the revision, we add the description of possible causes as follows,

"Li and Feng (2012) suggested that the different patterns of the Hadley circulation anomalies between the CP and EP El Niños are associated with the contrasting underlying thermal structure changes because the maximum of the zonal-mean SST anomalies is moved northward to about 10°N in the CP event relative to the maximum around the equator in the EP event."

13. Line 312: This should be "super" not "supper".

The error is corrected in the revision.

14. Line 324: What distinguishes the 09/10 event from the other El Niño events, including the other CP event, to cause such a large anomaly in the monsoon index? Can you explain/postulate why the CP El Niño tend to have a higher monsoon-related anomaly?

By using the reanalysis and ONI data in 1979-2019, we investigate the composite EP and CP El Niños from the 6 EP El Niño events (1982/1983, 1986/1987, 1991/1992, 1997/1998, 2006/2007 and 2015/2016 winters) and 5 CP El Niño events (1994/1995, 2002/2003, 2004/2005, 2009/2010 and 2018/2019 winters), as shown in Fig. 10 (presented in the revision).

[Figure]

**Figure 10.** Anomalies of (a, b) CWV and velocity potential and divergent wind at 850 hPa in (c, d) total, (e, f) Walker and (g, h) monsoon circulations for composite EP and CP El Niños derived from reanalysis data. The left and right columns correspond to the composite EP and CP El Niños, respectively. The shading and arrow in Fig. 10 (c-h) denote the velocity potential and divergent wind anomalies, respectively. The red and blue plus denotes the five radiosonde stations.

The composite events show that at the five radiosonde stations, the index of the Walker circulation anomaly accounts for about 75.8% (47.8%) of the total anomaly index in EP (CP) El Niño, while for the monsoon circulation, the anomaly index of 6.16 (4.66) units contributes to 49.6% (18.4%) of the total anomaly index in CP (EP) El Niño. Hence, the CP event can generally cause a large monsoon circulation anomaly. The 2009/10 CP event is the strongest CP El Niño from the 1980s, as observed by satellite measurements (Lee and Mcphaden, 2010). In this way, it is possible for the strongest CP El Niño in the winter of 2009/10 to cause a very strong monsoon circulation anomaly.

The whole description of the composite EP and CP El Niños, please see Section 4.3.

And according to your suggestion, we add the corresponding explanation in the revision as,

"In addition, at the radiosonde sites, the CP El Niño can generally cause an intense monsoon circulation anomaly, which is comparable to and even larger than the Walker circulation anomaly, thus the CP El Niño in the winter of 2009/10 may induce a quite strong monsoon circulation anomaly now that the 2009/10 event is the strongest CP El Niño from the 1980s, as observed by satellite (Lee and Mcphaden, 2010)."

15. Line 331: This is a dramatic understatement. The difference between the radiosondes and reanalysis is almost a factor of 2 different for one of the years, and about a factor of 5 for another. That's half of the data you show. This sentence needs to be reworded. Also, your assertion the CWV increases with increasing index only really applies to the sonde data. You need to qualify this statement.

As you pointed out, the difference between the radiosondes and reanalysis looks very large, which may cause the reader to think a very remarkable difference in the water vapor between the radiosondes and reanalysis.

In fact, the CWV is 44.87 (44.10), 43.06 (40.23), 41.16 (39.83) and 44.07 (42.87) kg m$^{-2}$ in the radiosonde (reanalysis) data in the 2006/07, 2009/10, 2015/16 and 2018/19 events, respectively. The relative difference is very small with only 1.7% in 2006/07, and 6.6% in 2009/10.

The average is 45.61 (44.17) kg m$^{-2}$ in the radiosonde (reanalysis) data from 2005 to 2019, thus the CWV anomaly in the radiosonde (reanalysis) data is -0.74 (-0.07) and -2.55 (-3.94) kg m$^{-2}$ in the 2006/07 and 2009/10 events. This causes that the CWV anomaly looks very large as the factors of 10.6 and 1.5 times, especially in 2006/07, but the differences of both the CWV and CWV anomaly

values in 2006/07 are very small between the radiosonde and reanalysis.

Qualitatively, the total circulation anomaly is large in 2009/10 and 2015/16, and small in 2006/07, and then corresponding CWV anomalies in both the radiosonde and reanalysis are large in 2009/10 and 2015/16, and small in 2006/07, respectively.

According to your suggestion, "Although there is some difference in the intensity of the CWV anomaly between the reanalysis and radiosonde data, both of them increase with the increasing index of the total circulation anomaly." is reworded as

"It can be seen from Fig. 9 that qualitatively, the CWV anomalies in the reanalysis and radiosonde data increase with the increasing index of the total circulation anomaly."

16. Line 352: As discussed above, a much more thorough analysis of the accuracy of the water vapor is warranted than what is described here and in Section 5.

According to your suggestion, we carry out a thorough comparison of water vapor among the reanalysis data and the radiosonde and satellite observations, as presented in our response to Major Comments 1, and in Section 6 of discussion in the revised manuscript.

Here, we rewrite the sentence of "However, in the first two events, there is a distinct difference between the reanalysis and radiosonde data. At least in the 2009/10 winter, we speculate that the reanalysis data may underestimate the tropospheric water vapor over the five stations, which can further be confirmed by the changes in the cloud and OLR data." to be

"However, in the first two events, there is a distinct difference of the CWV anomaly between the reanalysis and radiosonde data, and we will discuss the discrepancy in detail below."

17. Line 374: It would make far more sense just to evaluate the water vapor product with actual water vapor observations than the hand waving argument used here. Also, what implications does this have for the rest of the analysis, if you aren't confident in the accuracy of the water vapor product?

In the revised manuscript, we add a thorough comparison of water vapor among the reanalysis data and the radiosonde and satellite observations to evaluate the water vapor product, as presented in our response to Major Comments 1, and in Section 6 of discussion.

In the revised manuscript, the sentence "Therefore, this supports the radiosonde observation and our

suggestion that the reanalysis data underestimates the tropospheric water vapor over the radiosonde stations in the 2009/10 winter." is rewritten as

"Therefore, this supports the radiosonde observation that the water vapor over the radiosonde stations in the 2009/10 winter may be moister than in the reanalysis."

18. Figure 4: The plusses denoting the radiosonde sites aren't legible on the map. Please change the color.

According to your suggestion, the blue plusses replaced with the red plusses overlay on the dark blue background to represent the radiosonde sites in Fig.4.

**References**

[revised manuscript text omitted]

---

## Author Comment (AC2)

**Response to reviewer 2:**

We deeply appreciate your valuable suggestions on our manuscript, which has contributed greatly to the improvement of our manuscript.

We have carefully read these comments and suggestions. According to your suggestions, we revised our manuscript, and presented a point-to-point reply to your comments. The comments from the referee are highlighted in blue. Our responses and a brief summary of related changes to the manuscript are given below. Meanwhile, we also provided a revision with tracking the changes.

Review of Du et al. "Water vapor anomaly over the tropical western Pacific in El Nino winters from radiosonde and satellite observations". This study uses a combination of radiosonde, reanalysis, and satellite measurements to better understand how tropospheric water vapor changes over the tropical western Pacific during boreal winter. It is very helpful that this works seeks to pull apart the contributions from the Hadley, Walker, and monsoon circulations in these water vapor anomalies. It is a useful analysis but the difficult part is that over this 15 years there are only a few events and some are different types so there is a real struggle to determine what responses are most robust. The grammar is not clear in many places in the text and could use additional work. I have some suggestions for the authors to consider in order to recommend publication but it should be of interest to the ACP readership.

We thank you for your encouragement, and do our best to improve our manuscript according to the review and editor valuable suggestion.

**Comments:**

1. The title indicates using radiosonde and satellite observations but the abstract talks about radiosonde and reanalysis with a brief mention of satellite observations at the end, maybe adding reanalysis to the title would be best or include some big picture details related to the satellite analysis in the abstract.

According to your suggestion, the title is changed as "Water vapor anomaly over the tropical western Pacific in El Niño winters from radiosonde and satellite observations and ERA5 reanalysis data" in the revision.

In the abstract, we add the description of "In addition, a detailed comparison of water vapor in the reanalysis, radiosonde and satellite data shows a fine confidence level of the datasets".

2. The authors present a nice set of measurements that are included in the analysis. As I mentioned the main struggle is that in 15 years there are just a handful of events and some are different types. It is more difficult to gauge the robustness in the details because of this. The use of reanalysis does go back to much earlier time periods.

According to your suggestion, we go back to investigate the reanalysis data and the El Niño events from 1979. There exist 6 EP El Niño events (1982/1983, 1986/1987, 1991/1992, 1997/1998, 2006/2007 and 2015/2016 winters) and 5 CP El Niño events (1994/1995, 2002/2003, 2004/2005, 2009/2010 and 2018/2019 winters) in 1979-2019. Similar to Fig. 10 in the revision (also Fig. 9 in the previous manuscript), Fig. R21 (only in this Response) shows the indices of the Hadley, Walker, monsoon and total tropical circulation anomalies in the velocity potential at 850 hPa and CWV anomalies averaged at the five radiosonde stations in the 11 EP El Niño winters during 1979-2019.

**Figure R21.** (Left) Indices of (red) Hadley, (yellow) Walker, (blue) monsoon and (orange) total circulation anomalies and (right) CWV anomalies derived from reanalysis data at five radiosonde stations in 11 El Niño winters during 1979-2019.

Overall, there are very strong three EP El Niño events in the 1982/1983, 1991/1992 and1997/1998 winters, corresponding to strong total circulation anomaly index and CWV anomaly. From Fig. R21, we can draw three conclusions: 1) "The variability of the Hadley circulation is quite small and has little influence on the water vapor anomaly" (in the manuscript) in the 11 events; 2) "The anomaly of the Walker circulation makes a considerable contribution to the total anomaly in all El

Niño winters, especially in the eastern-Pacific (EP) El Niño events" (in the manuscript), which is very obvious in the 6 winters of 1982/1983, 1986/1987, 1991/1992, 1997/1998, 2006/2007 and 2015/2016 EP events; and 3) "The monsoon circulation shows a remarkable change from one event to another" (in the manuscript). Therefore, the main results in the manuscript can further be demonstrated by the 41 year data, in other words, the data spanning 40 years presents the consistent results with the 15 year data in the manuscript.

The considerable contribution from the Walker circulation anomaly is understandable since the Walker circulation located over the Pacific Ocean is directly related to water vapor transport over the Pacific. The remarkable change of the monsoon circulation from one event to another is also reasonable because the monsoon circulation can be significantly affected by more factors, for example, sea-land heat contrast, SST warming position, seasonal movement of the subtropical high, and even possible interoceanic coupling and linkage. However, the specific causes of the monsoon circulation anomaly in different events, particularly in different CP events, needs to be explored deeply and systematically, and please permit us to present the definite reason in the next work after compare and analyze each event in detail.

Considering your valuable suggestion, we add the investigation of composite EP and CP El Niños from the 6 EP El Niño events and 5 CP El Niño events based on the 41 year reanalysis data. The main results shown in the composite EP and CP events, such as little variability of the Hadley circulation, major contribution of the Walker circulation anomaly, especially in the EP events, and relatively important role of the monsoon circulation in the CP event compared with in EP event, are in agreement with the case study. In the revision, we add Fig. 10 and corresponding description I Section 4.3, as follows,

"In order to obtain the general features of water vapor and circulation anomalies in the EP and CP El Niño events, we extend the reanalysis data back to 1979 to examine two types of composite El Niño events. There are six EP El Niño events in the winters of 1982/83, 1986/87, 1991/92, 1997/98, 2006/07 and 2015/16, and five CP El Niño events in the 1994/95, 2002/03, 2004/05, 2009/10 and 2018/19 winters for 41 years from 1979 to 2019, which are averaged as the composite EP and CP El Niños, respectively. We calculate the CWV anomalies in the two composite events based the climatic mean CWV in 41 winters, and the corresponding velocity potential and divergent wind

anomalies of the Walker, monsoon and total circulations from the reanalysis horizontal wind at 850 hPa, which are shown in Fig. 10. The Hadley circulation anomaly (not presented) is very small, and its patterns in the composite EP and CP El Niños are also analogous to those in the EP and CP events shown in Fig. 6, respectively. On the whole, Figure 10 illustrates that the total circulation anomaly is stronger in EP event than in CP event, and then the CWV anomaly is larger in EP event relative to that in CP event. The Walker circulation plays an important role in the total circulation anomaly, especially in EP El Niño. Despite significant variability from one event to another, the monsoon circulation anomaly has not only a larger proportion of the total anomaly but also slightly higher intensity in CP El Niño than in EP El Niño. At the five radiosonde stations, the composite events indicate that the CWV anomaly is about -4.36 and -1.74 kg m-2 in EP and CP El Niños, respectively. The index of the Walker circulation anomaly accounts for about 75.8% (47.8%) of the total anomaly index in EP (CP) El Niño, while for the monsoon circulation, the anomaly index of 6.16 (4.66) units contributes to 49.6% (18.4%) of the total anomaly index in CP (EP) El Niño. Therefore, the relative importance of the Hadley, Walker and monsoon circulation anomalies in the composite El Niños is roughly in accord with that in the case study above."